# Auditory sensory deprivation induced by noise exposure exacerbates cognitive decline in a mouse model of Alzheimer's disease

Fabiola Paciello[1,2], Marco Rinaudo[2], Valentina Longo[2], Sara Cocco[2], Giulia Conforto[2], Anna Pisani[3], Maria Vittoria Podda[1,2], Anna Rita Fetoni[1,3]*, Gaetano Paludetti[1,3], Claudio Grassi[1,2]

[1]Fondazione Policlinico Universitario A. Gemelli IRCCS, Rome, Italy; [2]Department of Neuroscience, Università Cattolica del Sacro Cuore, Rome, Italy; [3]Department of Otolaryngology Head and Neck Surgery, Università Cattolica del Sacro Cuore, Rome, Italy

**Abstract** Although association between hearing impairment and dementia has been widely documented by epidemiological studies, the role of auditory sensory deprivation in cognitive decline remains to be fully understood. To address this issue we investigated the impact of hearing loss on the onset and time-course of cognitive decline in an animal model of Alzheimer's disease (AD), that is the 3×Tg-AD mice and the underlying mechanisms. We found that hearing loss induced by noise exposure in the 3×Tg-AD mice before the phenotype is manifested caused persistent synaptic and morphological alterations in the auditory cortex. This was associated with earlier hippocampal dysfunction, increased tau phosphorylation, neuroinflammation, and redox imbalance, along with anticipated memory deficits compared to the expected time-course of the neurodegenerative phenotype. Our data suggest that a mouse model of AD is more vulnerable to central damage induced by hearing loss and shows reduced ability to counteract noise-induced detrimental effects, which accelerates the neurodegenerative disease onset.

*For correspondence:
annarita.fetoni@unicatt.it

## Introduction

Recent epidemiological evidence suggests a strong association between hearing loss and cognitive decline (*Gallacher et al., 2012*; *Thomson et al., 2017*; *Livingston et al., 2020*; *Loughrey et al., 2018*; *Liu and Lee, 2019*). Specifically, impairments in peripheral and central auditory structures have been linked to incidence and acceleration of cognitive deficits (*Bernabei et al., 2014*; *Amieva et al., 2015*; *Fortunato et al., 2016*; *Deal et al., 2017*) as well as to increased risk for the onset of neurodegenerative disorders including Alzheimer's disease (AD) (*Gates et al., 2002*; *Panza et al., 2015*; *Taljaard et al., 2016*; *Zheng et al., 2017*; *Shen et al., 2018*). Accordingly, it has been shown that for every 10 dB increase in hearing loss, there is a 20 % increased risk of developing dementia (*Lin et al., 2011*). Substantiating such correlation would have significant implications for prevention and treatment of dementia. Indeed, while it is difficult to counteract neurodegeneration, hearing loss can be considered as a modifiable risk factor, given that it could be widely treated with hearing aids or cochlear implants. Therefore, it is mandatory to clarify the mechanisms linking hearing loss to dementia. Several hypotheses have been proposed to explain the relationship between auditory sensory deprivation and cognitive impairment (*Griffiths et al., 2020*; *Johnson et al., 2021*; *Slade et al., 2020*), but the nature of such association remains controversial.

It has been reported that hearing loss causes a cascade of changes in the main auditory pathway and in non-lemniscal brain regions, such as the hippocampus (*Manikandan et al., 2006*; *Goble et al., 2009*; *Cui et al., 2018*; *Nadhimi and Llano, 2021*), a brain structure involved in memory and severely impaired in cognitive decline and AD. Sensory disruption due to damage of the organ of Corti may trigger central mechanisms of homeostatic plasticity (*Rauschecker, 1999*; *Syka, 2002*; *Caspary et al., 2008*; *Wang et al., 2011*; *Yang et al., 2011*) and changes in excitatory, inhibitory, and neuromodulatory networks along the central auditory pathway have been described (*Liberman and Kiang, 1978*; *Abbott et al., 1999*; *Milbrandt et al., 2000*; *Salvi et al., 2000*; *Richardson et al., 2012*; *Engineer et al., 2013*). In previous studies we showed that exposure to loud sounds led to structural plasticity changes in central auditory structures, resulting in decreased spine density and altered dendritic complexity in pyramidal neurons of layer II/III of the auditory cortex (ACx) (*Fetoni et al., 2013*; *Fetoni et al., 2015*; *Paciello et al., 2018*). It is also known that neurons in the hippocampus respond to acoustic stimuli (*Moita et al., 2003*; *Xiao et al., 2018*); indeed, auditory potentials can be evoked in this structure (*Bickford-Wimer et al., 1990*; *Moxon et al., 1999*). Moreover, data from animal models suggest that hearing loss can affect hippocampal functions by altering neurotransmitter levels (*Cui et al., 2009*; *Chengzhi et al., 2011*; *Cui and Li, 2013*; *Beckmann et al., 2020*) and by decreasing neurogenesis (*Tao et al., 2015*; *Liu et al., 2015*; *Kurioka et al., 2021*). However, to our knowledge, no studies have yet characterized the impact of auditory sensory deprivation on the onset and time-course of cognitive decline in AD and the underlying mechanisms. To this aim, we used 3×Tg AD mice, a common experimental model of AD, which experienced hearing loss induced by noise exposure at an age of 2 months, when the AD phenotype is not manifested yet. Auditory and hippocampal functions were then investigated over time to establish whether and how auditory sensory deprivation could accelerate and/or worsen AD-associated cognitive impairment.

## Results

## Noise exposure induces hearing loss in both 3×Tg-AD and wild-type mice

Our first step was to characterize hearing loss induced by noise at different time points in both 3×Tg -AD and wild-type (WT) mice. Thereinafter mice exposed to noise will be referred to as 'AD-NE' and 'WT-NE', whereas normal hearing mice not subjected to noise as 'AD-NN' and 'WT-NN'. Animals were exposed to noise at 2 months of age (M) and subsequently analyzed at 3 M (corresponding to

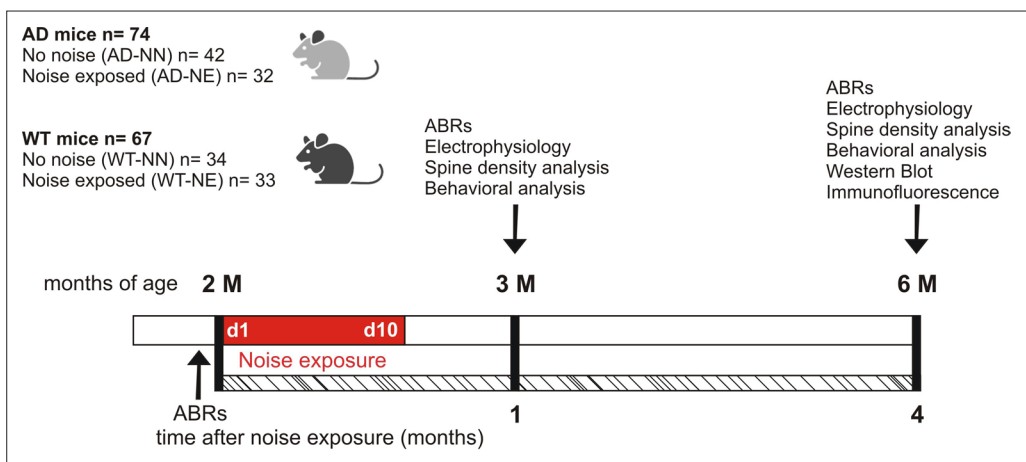

**Figure 1.** Schematic representation of the experimental design and time schedule of the protocols. Wild-type (WT) and 3×Tg Alzheimer's disease (AD) animals of 2 months of age (M) at the beginning of the study were randomly assigned to no noise (NN) or noise-exposed (NE) groups. Baseline hearing thresholds were evaluated the day before the exposure to repeated noise sessions lasting 10 consecutive days (d1–d10). After 1 and 4 months from the onset of trauma sessions, when the mice aged 3 and 6 M, behavioral, morphological (spine density), electrophysiological, and molecular (Western blot and immunofluorescence) evaluations were performed. ABRs: auditory brainstem responses.

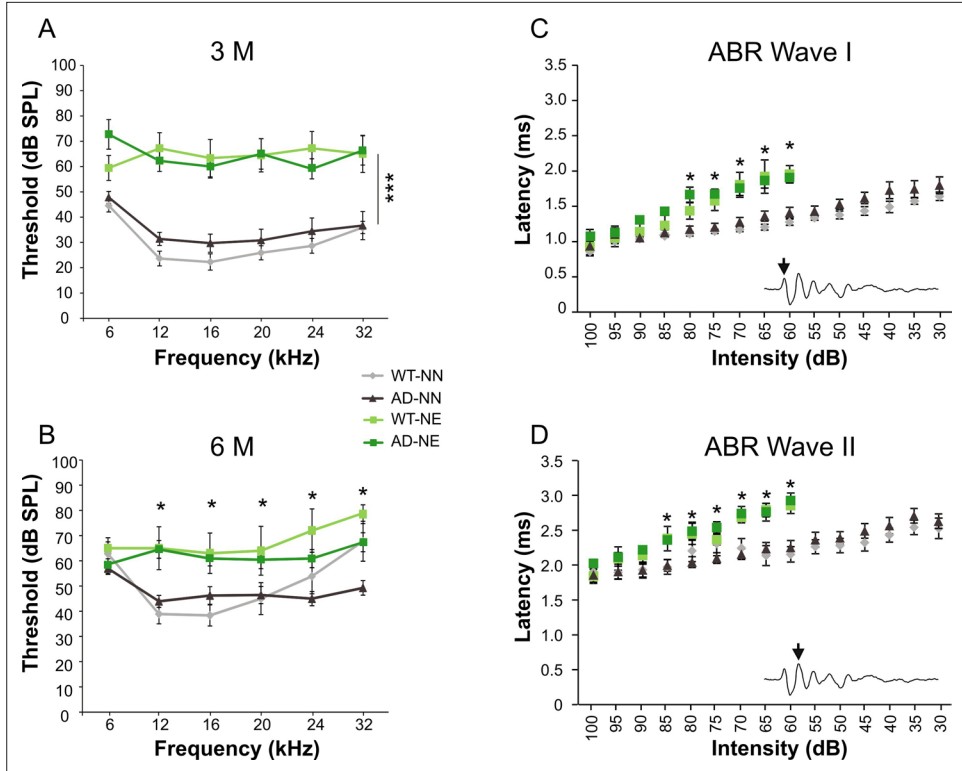

**Figure 2.** Auditory threshold evaluations after noise exposure. (A–B) Graphs show ABR averaged threshold values (± SEM) measured in not-exposed (NN) and noise-exposed (NE) wild-type (WT) and in Alzheimer's disease (AD) mice. Auditory thresholds were similar in WT-NN and AD-NN mice at both 3 (**A**) and 6 (**D**) months of age (WT-NN 3 M n = 11; AD-NN 3 M n = 12; WT-NN 6 M n = 9; AD-NN 6 M n = 13). Repeated noise exposure induced a threshold elevation of about 30–40 dB after 1 month from the onset of trauma sessions in both strains (WT-NE 3 M n = 8; AD-NE 3 M n = 7; three-way ANOVA, Tukey's post hoc test, WT-NE vs. WT-NN, p = 0.0001; AD-NE vs. AD-NN, p = 0.0002) and no recovery in threshold and threshold shift values was observed 4 months after noise exposure, corresponding to 6 M, compared with age-matched not-exposed animals (WT-NE 6 M n = 7; AD-NE 6 M n = 11; three-way ANOVA, Tukey's post hoc test, WT-NN vs. WT-NE, p = 0.017; AD-NN vs. AD-NE, p = 0.003). (C–D) Graphs show Wave I (**C**) and II (**D**) latency-intensity functions across animals of 3 M in response to 16 kHz tone bursts (WT-NN n = 8; AD-NN n = 10; WT-NE n = 7; AD-NE n = 7). Asterisks indicate significant differences between groups (*p < 0.05; ***p < 0.001).

The online version of this article includes the following figure supplement(s) for figure 2:

**Source data 1.** Numerical source data from *Figure 2*.

1 month after noise exposure) and at 6 M (corresponding to 4 months after noise exposure). Experimental design and protocol timeline of experiments are summarized in *Figure 1*.

The hearing loss was evaluated by auditory brainstem recording (ABR) threshold measurements. Baseline mean ABR thresholds ranged from 40 to 20 dB sound pressure level (SPL) across frequencies, with no significant differences between 3×Tg -AD and WT mice (*Figure 2A*; AD-NN n = 12; WT-NN n = 11; three-way ANOVA, Tukey's post hoc test, p = 0.80). In line with our previous reports (*Fetoni et al., 2013*; *Paciello et al., 2018*), repeated noise exposure worsened threshold pattern, reaching a threshold increased value of about 40 dB 1 month after noise exposure (3 M, *Figure 2A and C*; WT-NE n = 8, AD-NE n = 7; three-way ANOVA, Tukey's post hoc test, WT-NE vs. WT-NN, p = 0.0001; AD-NE vs. AD-NN, p = 0.0002). The trend of threshold increase was similar across mid- and high frequencies and no significant differences were observed between 3×Tg -AD and WT mice, indicating that both strains show a similar degree of susceptibility to hearing loss induced by noise. At 6 M, both WT-NN and AD-NN showed a slight increase (about 20 dB) in auditory threshold values (*Figure 2B*) compared to younger mice, probably as a consequence of physiological age-related worsening of cochlear function. Of note, when comparing noise-exposed 6 M animals with age-matched not-exposed animals, a significant increase in auditory threshold was observed, specifically in the range of mid-frequencies,

indicating that noise can induce a permanent hearing loss over time (*Figure 2B*; WT-NN n = 9, AD-NN n = 13; WT-NE n = 7, AD-NE n = 11; three-way ANOVA, Tukey's post hoc test, WT-NN vs. WT-NE, p = 0.017; AD-NN vs. AD-NE, p = 0.003). In fact, an auditory threshold worsening of about 20–25 dB was still detectable in noise-exposed mice, compared to age-matched not-exposed animals, in both 3×Tg -AD and WT mice with no significant differences between strains.

Analysis of latency-intensity curves showed that noise exposure significantly increased latency of ABR Waves I and II at different intensities (*Figure 2C–D*; WT- NN n = 8, AD-NN n = 10; WT-NE n = 7, AD-NE n = 7; Student's t-test) in both WT and 3×Tg -AD mice. These findings suggest that noise exposure induced an impairment of the number of neural unit firing.

Collectively, these data indicate that noise exposure can induce a hearing loss that persists over time and similarly affects cochlear function in both 3×Tg -AD and WT mice.

## Synaptic function and spine density in neurons of layer II/III of ACx are most severely affected by hearing loss in 3×Tg-AD mice

In order to evaluate the effect of sensory deprivation induced by noise exposure in the ACx, we studied field excitatory post-synaptic potentials (fEPSPs) in ACx layer II/III following stimulation of local connections in both 3×Tg -AD and WT mice 1 and 4 months after acoustic trauma, to assess early and long-lasting changes related to cochlear damage, respectively. As expected on the basis of our previous findings (*Paciello et al., 2018*), 1 month after the onset of trauma sessions, comparison of the input/output (I/O) curves, obtained by plotting EPSP amplitude against stimulus intensities, showed that fEPSPs were significantly smaller in animals subjected to noise compared to those not exposed, with similar results in WT (*Figure 3A*; n = 17 slices from three WT-NE and n = 11 slices from five WT-NN; two-way ANOVA, Tukey's post hoc test, $F_{(1,126)}$ = 59.5, p < 0.001) and 3×Tg -AD mice (*Figure 3B*; n = 13 slices from five AD-NE and n = 15 slices from six AD-NN; two-way ANOVA, Tukey's post hoc test, $F_{(1,126)}$ = 15.302, p < 0.001).

Interestingly, at 6 M, no significant differences between WT-NN and WT-NE animals (*Figure 3C*; n = 20 slices from eight WT-NE and n = 21 slices from seven WT-NN mice; two-way ANOVA, Tukey's post hoc test, $F_{(1,117)}$ = 0.084, p = 0.773) were observed. Conversely, at 6 M, decreased response (of about 30%) was found in AD-NE compared to age-matched not-exposed animals (*Figure 3D* and *Figure 3— figure supplement 1*; n = 14 slices from five AD-NE and n = 12 slices from five AD-NN mice; two-way ANOVA, Tukey's post hoc test, $F_{(1,72)}$ = 12.53, p < 0.001), suggesting that in this mouse strain auditory cortex was most severely affected by noise.

To further assess the effect of hearing loss on glutamatergic transmission, phosphorylation of AMPA receptor (AMPAR) GluA1 subunit at Ser845 (pGluA1$^{Ser845}$) was also evaluated in our experimental conditions because of its role in AMPAR function, trafficking, and channel conductance (*Derkach et al., 1999*; *Lee et al., 2000*; *Suzuki et al., 2005*; *Havekes et al., 2007*; *Man et al., 2007*). Although no difference in pGluA1$^{Ser845}$ expression was found in ACx of of 3 M animals (*Figure 3—figure supplement 2*), our Western immunoblot analyses performed on ACx extracts from 6 M 3×Tg -AD and WT mice revealed that hearing loss significantly decreased the levels of pGluA1$^{Ser845}$ (by approximately 60%) in AD-NE mice (*Figure 3E*; n = 4 animals/group; Student's t-test, AD-NE vs. AD-NN, p = 0.0001), compared to age-matched not-exposed animals.

The effect of noise exposure on glutamatergic synapses was also evaluated at structural level by analyzing spine density in pyramidal neurons of ACx layer II/III. Consistent with our previous studies (*Fetoni et al., 2015*; *Paciello et al., 2018*), 1 month after noise exposure, the number of spines in apical and basal dendrites in both WT and 3× Tg AD animals was significantly reduced (*Figure 4— figure supplement 1*). In line with functional evaluations, analysis at the subsequent time point (i.e., 4 months after noise exposure) showed a decreased spine density in both apical and basal dendrites in neurons of 6 M AD-NE mice compared with age-matched AD-NN animals (*Figure 4B and D*; two-way ANOVA, Tukey's post hoc test, $F_{(1,142)}$ = 6.94, apical dendrites, n = 36 neurons from four AD-NN animals, n = 38 neurons from four AD-NE animals, p = 0.011; basal dendrites, n = 32 neurons from four AD-NN animals, n = 32 neurons from four AD-NE animals, p = 0.012). No significant differences in spine number were, instead, observed when comparing WT-NE and groups (*Figure 4A and C*; two-way ANOVA, Tukey's post hoc test, p = 0.88; n = 40 neurons from four WT-NN animals, n = 30 neurons from four WT-NE animals; basal dendrites, p = 0.70; n = 30 neurons from four WT-NN animals, n = 30 neurons from four WT-NE animals). In keeping with these data, Western blot analysis

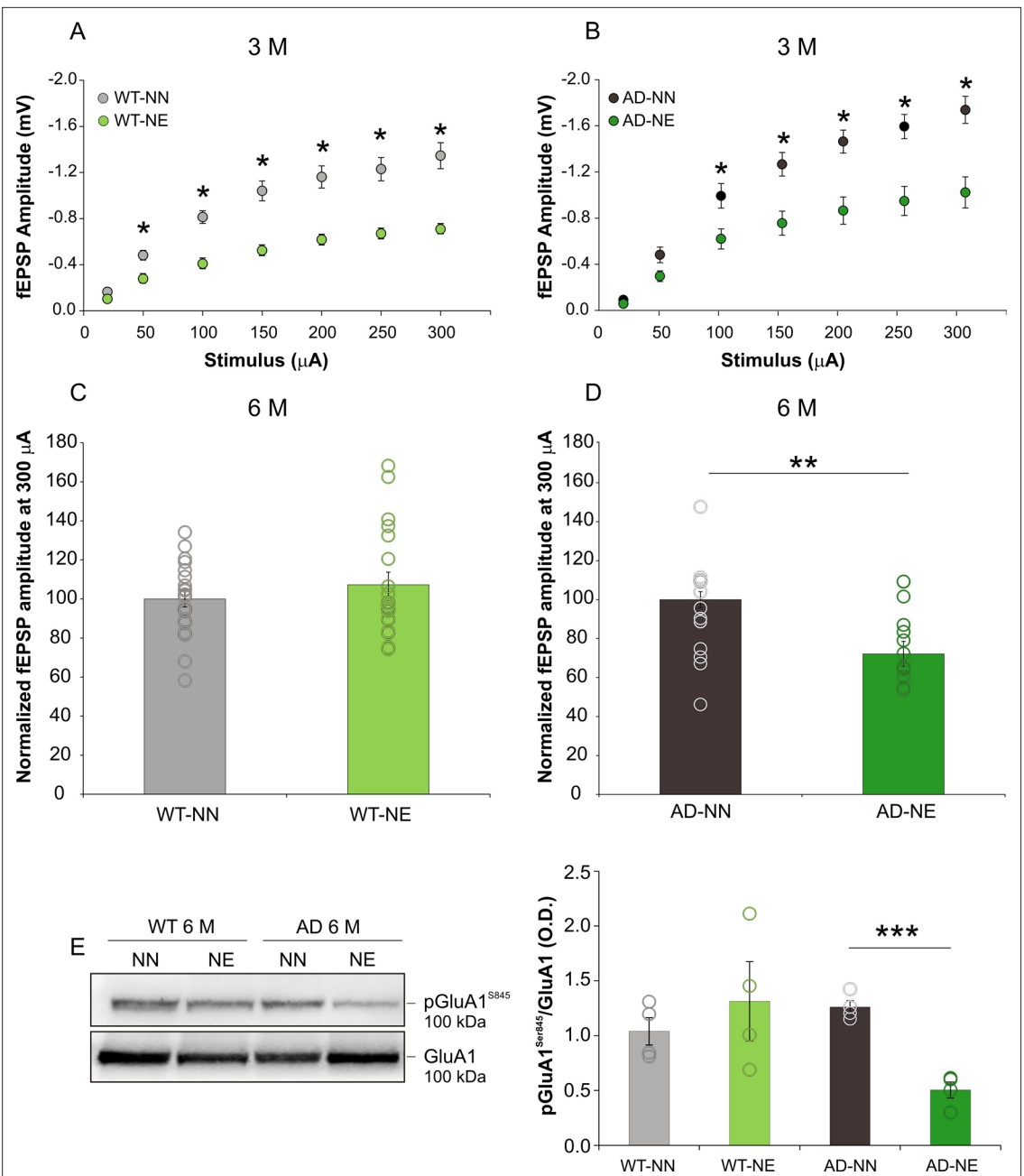

**Figure 3.** Hearing loss induces persistent synaptic dysfunctions in the ACx of 3×Tg Alzheimer's disease (AD) mice. (A–B) Field excitatory post-synaptic potential (fEPSP) amplitude following stimulation of afferent fibers in ACx layer II/III at increasing intensities are shown for slices obtained from 3 months of age (M) not-exposed (NN) and noise-exposed (NE) wild-type (WT) (**A**) and 3×Tg -AD mice (**B**). Statistical analysis by two-way ANOVA followed by Tukey's post hoc revealed significant differences between not-exposed (WT n = 11 slices from mice; AD n = 15 slices from six mice) and noise-exposed (WT n = 17 slices from three mice; AD n = 13 slices from five mice) groups of both strains (WT-NN vs. WT-NE, p < 0.001; AD-NN vs. AD-NE, p < 0.001). (C,D) fEPSP amplitude normalized to mean values obtained in 6 M WT-NN and AD-NN at 300 µA, taken as 100 % (C; WT-NN n = 21 slices from seven mice; WT-NE n = 20 slices from eight mice; D, AD-NN n = 12 slices from five mice; AD-NE n = 14 slices from five mice). Statistical analysis by two-way ANOVA revealed significant differences between AD-NN and AD-NE groups (p = 0.001). (E) Representative Western immunoblot revealing decreased pGluA1Ser845 in the ACx of AD mice exposed to noise (6 M) compared with age-matched not-exposed group. Bar graph shows results of densitometric analyses on all samples (n = 4 mice for each group; Student's t-test, p = 0.071 WT-NE vs. WT-NN; p = 0.0001 AD-NE vs. AD-NN) normalized to the corresponding total protein levels (GluA1). Data are expressed as mean ± SEM. Asterisks indicate significant differences between groups (*p < 0.05; **p < 0.01;***p < 0.001).

The online version of this article includes the following figure supplement(s) for figure 3:

**Source data 1.** Numerical source data from *Figure 3*.

*Figure 3 continued on next page*

*Figure 3 continued*

**Figure supplement 1.** (A–B) Field excitatory post-synaptic potentials (fEPSP) amplitude following stimulation of afferent fibers in ACx layer II/III at increasing intensities are shown for slices obtained from 6 months of age (M) not-exposed (NN) and noise-exposed (NE) wild-type (WT) (**A**) and Alzheimer's disease (AD) mice (**B**).

**Figure supplement 2.** Representative Western immunoblot revealing pGluA1$^{Ser845}$ expression in the ACx of 3 months of age (M) animals.

showed significantly lower levels of PSD-95 in ACx of 3×Tg -AD mice and not in WT exposed to noise compared with their respective age-matched not-exposed animals (***Figure 4E and F***; n = 4 animals/ group; Student's t-test, AD-NE vs. AD-NN mice p = 0.025).

Collectively, these data suggest that 3×Tg -AD mice are most vulnerable to central damage induced by hearing loss as they cannot recover over time functional and morphological alterations induced by sensory deprivation in the ACx.

## Auditory sensory deprivation accelerates hippocampal dysfunction and memory deficits in 3×Tg-AD mice

Having established the long-lasting detrimental effects of hearing loss on ACx of 3×Tg -AD mice, we asked whether ACx damage impinged on hippocampal function, contributing to cognitive decline. Indeed, several evidences suggest that cortical and cognitive adaptation to the loss of a sensory modality impacts on information processing ability of cortical structures, and this, in turn, compromises the ability of the hippocampus to reliably and effectively encode and store sensory experience (***Feldmann et al., 2019***; ***Beckmann et al., 2020***).

To address this issue, we first analyzed synaptic function in the hippocampus of 6 M animals, an age at which, as shown above, functional and morphological alterations in the ACx persists only in 3×Tg -AD and not in WT mice.

fEPSPs in the CA1 area were measured after stimulation of Schaffer collaterals at increasing stimulus intensities in slices from both non-exposed and noise-exposed 3×Tg -AD and WT mice. As shown in ***Figure 5***, noise affected basal synaptic transmission only in AD mice. In fact, fEPSP amplitudes were significantly reduced in AD-NE mice with respect to age-matched AD-NN animals (***Figure 5B***; n = 11 slices from four AD-NE mice and n = 12 slices from four AD-NN mice; two-way ANOVA, Tukey's post hoc test, $F_{(1,126)}$, p = 0.020). On the other hand, no significant differences were observed between noise-exposed and non-exposed animals of WT strain (***Figure 5A***; n = 12 slices from six WT-NE mice and n = 14 slices from seven WT-NN mice; two-way ANOVA, Tukey's post hoc test, $F_{(1,120)}$, p = 0.78).

In line with functional data, morphological analyses showed a significant decrease of spine density in CA1 and dentate gyrus (DG) of the hippocampus in AD-NE mice. As shown in ***Figure 6***, a decreased number of dendritic spines were observed in transgenic mice exposed to noise compared with age-matched not-exposed animals, both in apical and basal dendrites of CA1 pyramidal neurons (***Figure 6B and D***; two-way ANOVA, Tukey's post hoc test, $F_{(1,63)}$ = 41.66, apical dendrites, n = 35 neurons from four AD-NN animals, n = 30 neurons from four AD-NE animals, p = 0.0001; basal dendrites, n = 37 neurons from four AD-NN animals, n = 34 neurons from four AD-NE animals, p = 0.0009) and in dendrites of DG granular cells (***Figure 6F and H***; one-way ANOVA, $F_{(1,63)}$ = 11.51, n = 31 neurons from four AD-NN animals, n = 34 neurons from four AD-NE animals, p = 0.001). No significant differences were observed between noise-exposed and not-exposed mice in the WT group (***Figure 6A, C, E and G***; CA1, two-way ANOVA, Tukey's post hoc test, $F_{(1,59)}$ = 0.21, p = 0.64, apical dendrites, n = 41 neurons from four WT-NN animals, n = 30 neurons from four WT-NE animals; basal dendrites, n = 31 neurons from four WT-NN animals, n = 31 neurons from four WT-NE animals; DG, one-way ANOVA $F_{(1,78)}$ = 0.16, p = 0.68, n = 44 neurons from four WT-NN animals, n = 36 neurons from four WT-NE animals). Western blot analysis performed on hippocampal extracts from both 3×Tg -AD and WT animals corroborates this result, showing a significantly lower expression of PSD-95 in 3×Tg -AD animals exposed to noise compared with not-exposed animals (***Figure 6I and J***; n = 4 animals/group; Student's t-test, AD-NE vs. AD-NN mice, p = 0.028).

We next ascertained whether these functional and morphological alterations were associated to deficits in hippocampal-dependent memory, assessed by the novel object recognition (NOR) test, evaluating short-term (STM) and long-term memory (LTM). The NOR test is a widely used behavioral paradigm for the assessment of object-recognition memory, in which an animal has to discriminate a novel from an old object. In a first phase of the test, the animal is allowed to explore two objects and then,

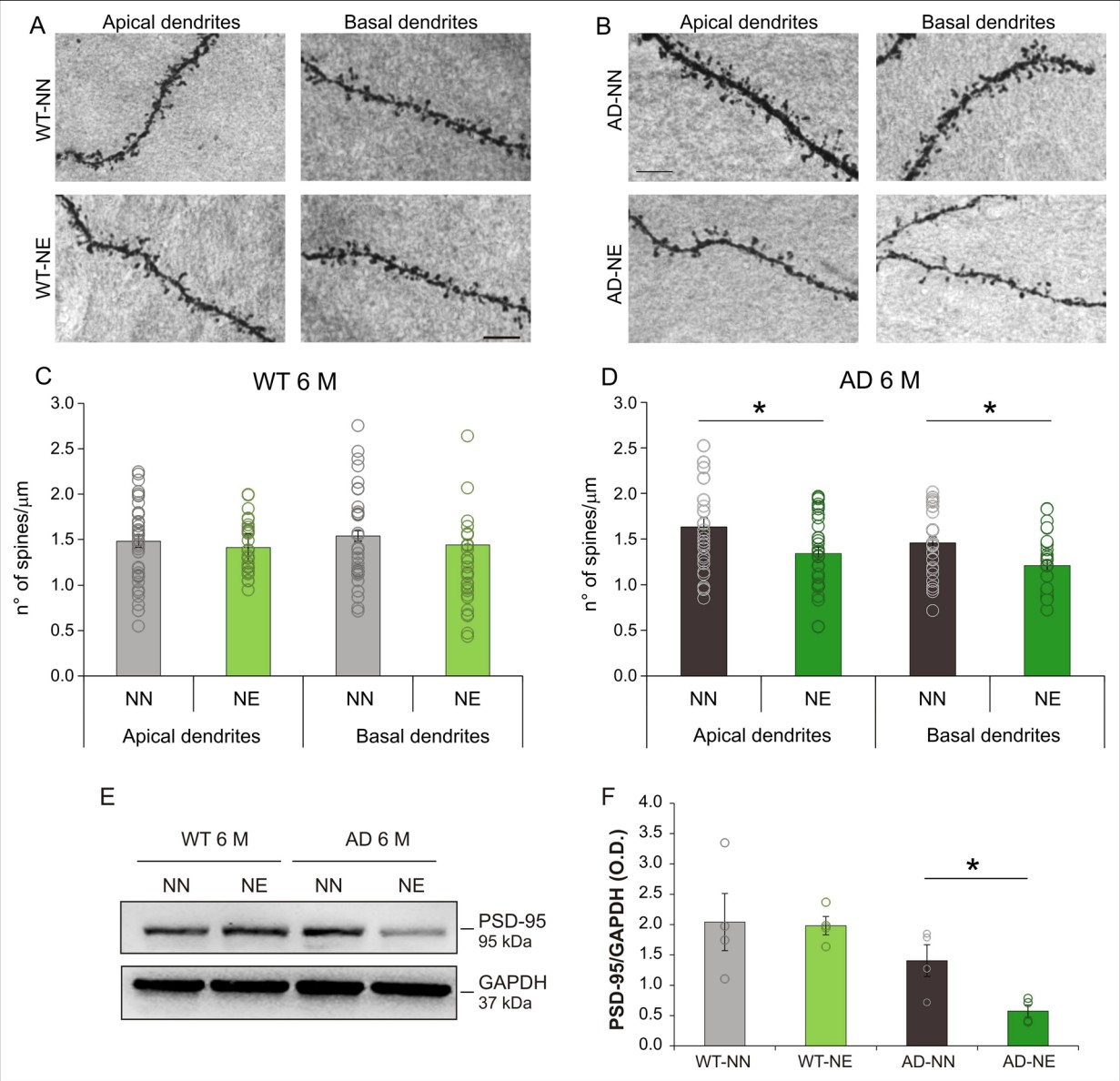

**Figure 4.** Hearing loss affects spine density in pyramidal neurons of ACx layer II/III in 3×Tg Alzheimer's disease (AD) mice. (A–B) Representative images of apical and basal dendrites of pyramidal neurons of layers II/III in WT-NN and WT-NE groups (left panels, **A**) and AD-NN and AD-NE groups (right panels, **B**) at 6 months of age (M). Scale bar: 10 μm. (C–D) Bar graphs showing mean values of spine density in apical and basal dendrites of neurons of layer II/III of the ACx of NN and NE wild-type (WT) (**C**) and AD mice (**D**) (n = at least 30 segments from 30 different neurons were analyzed from four animals/groups). The number of spines decreased significantly in AD-NE compared to AD-NN mice both in apical and in basal dendrites (D, two-way ANOVA, Tukey's post hoc test, apical dendrites p = 0.011; basal dendrites p = 0.012) whereas no differences between WT-NN and WT-NE groups were observed (C, two-way ANOVA, apical dendrites p = 0.88, basal dendrites p = 0.70). (E) Representative Western immunoblot revealing decreased PSD-95 expression in ACx of 6 M AD mice exposed to noise compared with age-matched not-exposed group. (F) Bar graph in the lower panel shows results of densitometric analyses on all samples (n = 4 mice for each group; Student's t-test, WT-NE vs. WT-NN, p = 0.90; AD-NE vs. AD-NN, p = 0.025) normalized to total protein levels (GAPDH). Data are expressed as mean ± SEM. Asterisks indicate significant differences between groups (*p < 0.05).

The online version of this article includes the following figure supplement(s) for figure 4:

**Source data 1.** Numerical source data from *Figure 4*.

**Figure supplement 1.** Spine density in pyramidal neurons of layer II/III in wild-type (WT) and Alzheimer's disease (AD) mice at 3 months of age (M).

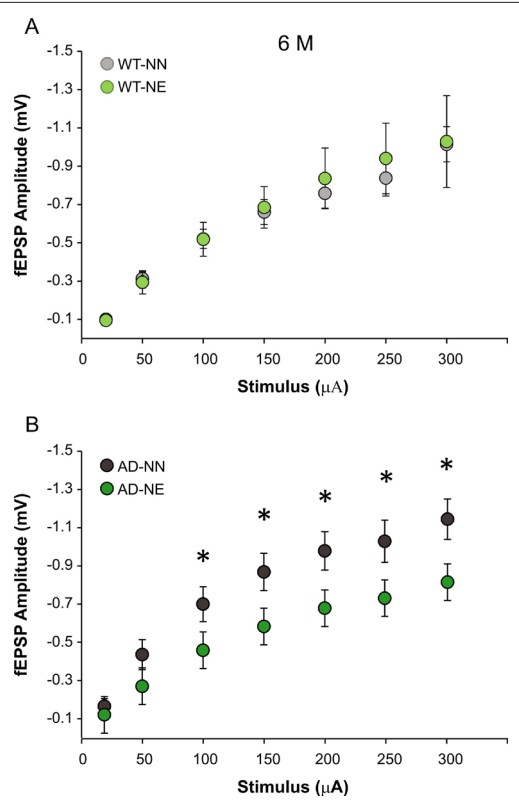

**Figure 5.** Hearing loss affects basal synaptic transmission in hippocampus of 3×Tg Alzheimer's disease (AD) mice. Field excitatory post-synaptic potential (fEPSP) amplitudes following stimulation of the Schaffer collaterals at increasing stimulus intensities in slices obtained from not-exposed and noise-exposed animals of both wild-type (WT) (**A**) and AD (**B**) groups at 6 months of age (M). Statistical analysis by two-way ANOVA followed by Tukey's post hoc revealed significant differences (p = 0.020) between AD-NN (n = 12 slices from four mice) and AD-NE (n = 11 slices from four mice) groups, whereas no significant differences (p = 0.78) were observed between WT-NN (n = 14 slices from seven mice) and WT-NE groups (n = 12 slices from six mice). Data are expressed as mean ± SEM. Asterisks indicate significant differences between groups (*p < 0.05).

The online version of this article includes the following figure supplement(s) for figure 5:

**Source data 1.** Numerical source data from *Figure 5*.

after a certain time interval, one of the objects is changed with a novel one. Depending on the time interval between the two sessions, it is possible to evaluate STM or LTM. Changes in the exploration time of the new and the old object are indicative of recognition memory, which depends on temporal lobe activity, specifically on the hippocampus (*Ennaceur and De Souza Silva, 2018*). We found that, at 3 M, 3×Tg -AD and WT mice exposed to noise did not exhibit any impairment in recognition memory compared to age-matched not-exposed animals (*Figure 7A and C*; *Table 1*). In particular, preference index was comparable in AD-NN and AD-NE groups. Interestingly, at 6 M memory performance was significantly altered in 3×Tg -AD exposed to noise compared with age-matched not-exposed animals, for both LTM and STM evaluations (*Figure 7B and D*; *Table 1*). Consistent with functional and morphological evaluations, no significant differences in recognition memory were found in WT animals exposed to noise compared to age-matched not-exposed mice (*Figure 7D*; *Table 1*).

Finally, differences in memory performance were independent of locomotor activity, as we did not observe any differences between NE and NN mice in total distance traveled (*Figure 7—figure supplement 1*).

Given that memory deficits in 3×Tg AD mice reportedly manifest at about 8 months (*Clinton et al., 2007*; *Stover et al., 2015*; *Cocco et al., 2020*), our data suggest that hearing loss accelerates memory impairment in this transgenic mouse model of AD.

Considering that memory performance by NOR test does not directly rely on auditory processing or auditory plasticity, we looked for a correlation between hearing loss and cognitive dysfunction, by including ABR measurements and NOR performance in an animal-by-animal study. Simple linear regression analysis across 6 M animals that underwent both ABR and NOR procedures (AD-NN n = 13; AD-NE n = 10; WT-NN n = 9; WT-NE n = 7) identified no relationship between ABR mean threshold and NOR performance in all experimental groups (*Figure 7—figure supplement 2*; AD-NN, $r^2$ = 0.0049, p = 0.467; AD-NE, $r^2$ = 0.0032, p = 0.878; WT-NN, $r^2$ = 0.187, p = 0.244; WT-NE, $r^2$ = 0.0009, p = 0.984). Similar results were obtained comparing NOR performance with mean spine density in ACx (*Figure 7—figure supplement 2*; AD-NN, $r^2$ = 0.240, p = 0.509; WT-NN, $r^2$ = 0.002, p = 0.953; WT-NE, $r^2$ = 0.206, p = 0.546, n = 4 animals/group).

However, of note, we found in AD-NE (n = 4) a significant statistical relationship (*Figure 7—figure supplement 2*; $r^2$ = 0.96, p = 0.019) between NOR and spine density data, suggesting that deficits in memory performance correlated with the central effects of hearing loss and altered structural plasticity of ACx circuitry.

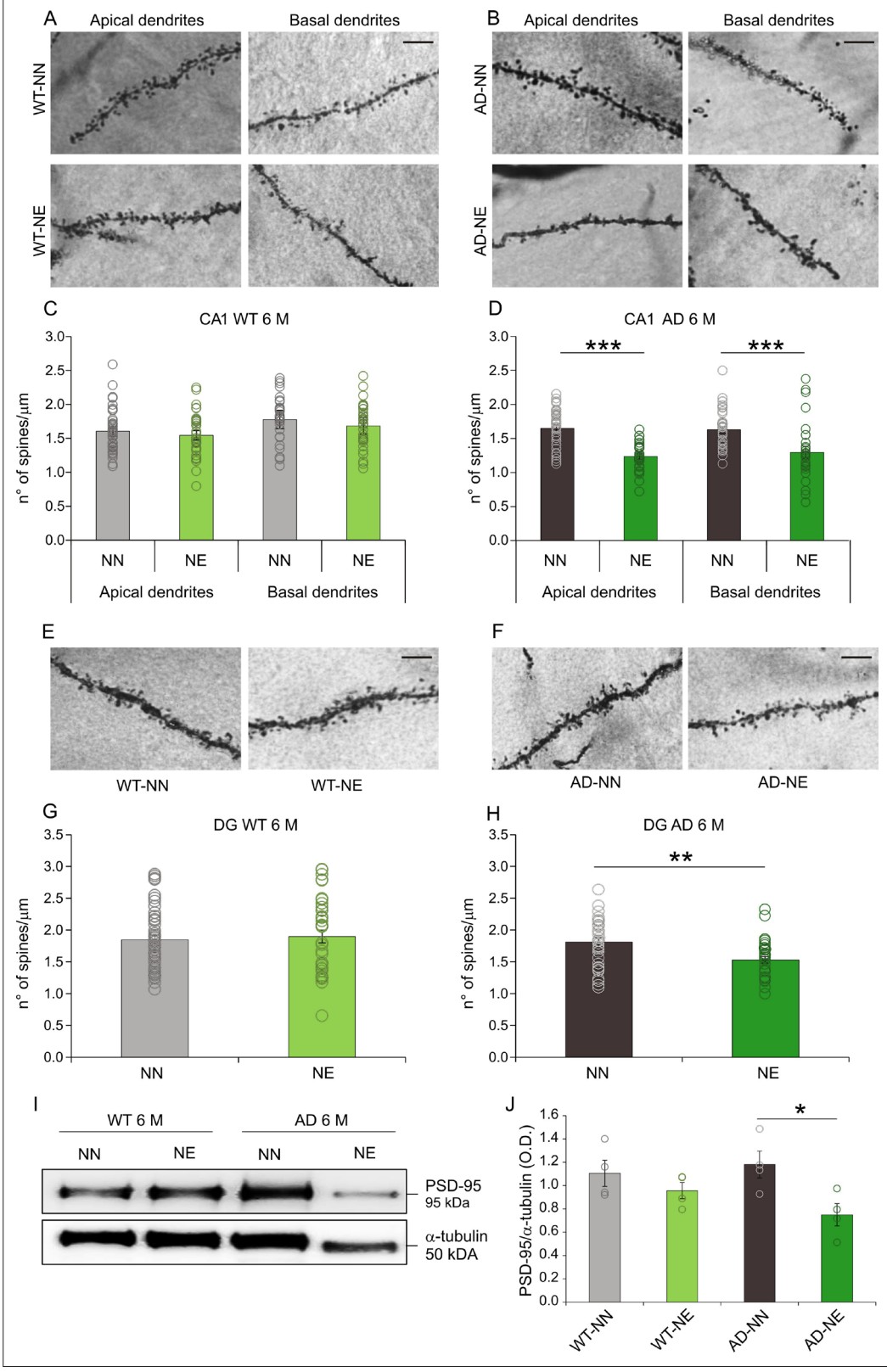

**Figure 6.** Decreased spine density is observed in the hippocampus of 3×Tg Alzheimer's disease (AD) mice exposed to noise. (A–B) Representative images of apical and basal dendrites of neurons in the hippocampal CA1 region in WT-NN and WT-NE groups (left panels, A) and in AD-NN and AD-NE groups (right panels, B). Scale bar: 10 μm. (C–D) Bar graphs showing mean values of spine density (mean ± SEM) in apical and basal dendrites of

*Figure 6 continued on next page*

*Figure 6 continued*

CA1 pyramidal neurons in NN and NE groups from WT (C) and AD mice (D) at 6 months of age (M). At least 30 segments from 30 different neurons were analyzed from four animals/groups. The number of spines decreased significantly in AD-NE compared to AD-NN group (D, two-way ANOVA, Tukey's post hoc test, apical dendrites p = 0.0001; basal dendrites p = 0.0009) whereas no differences between WT-NN and WT-NE groups were observed (C, two-way ANOVA, Tukey's post hoc test, p = 0.64). (E–F) Representative images of dentate gyrus (DG) neuron dendrites in WT-NN and WT-NE groups (left panels, E) and AD-NN and AD-NE groups (right panels, F). Scale bar: 10 μm. (G–H) Bar graphs showing mean values of spine density (mean ± SEM) in neurons of DG in NN and NE groups from WT mice (G) and AD mice (H) at 6 M. The number of spines decreased significantly in AD-NE compared to AD-NN mice (H, one-way ANOVA, p = 0.001) whereas no differences between WT-NN and WT-NE groups were observed (G, one-way ANOVA, p = 0.68). (I) Representative Western immunoblot revealing decreased PSD-95 expression in the hippocampus of AD exposed to noise (6 M) compared with age-matched not-exposed group. (J) Bar graph in the lower panel shows results of densitometric analyses on all samples (n = 4 mice for each group; Student's t-test, WT-NE vs. WT-NN, p = 0.30; AD-NE vs. AD-NN, p = 0.028) normalized to total protein levels (α-tubulin). Data are expressed as mean ± SEM. Asterisks indicate significant differences between groups (*p < 0.05; **p < 0.01; ***p < 0.001).

The online version of this article includes the following figure supplement(s) for figure 6:

**Source data 1.** Numerical source data from *Figure 6*.

Collectively, our results revealed that sensory deprivation induced by noise exposure affects hippocampal synaptic function, decreases spine density and accelerates memory deficits in 3×Tg -AD model.

## Molecular determinants of hippocampal dysfunction induced by hearing loss

Looking for molecular mechanisms underlying hearing loss-associated hippocampal functional and morphological alterations observed in 3×Tg -AD animals at 6 M, we focused on common hallmarks of neurodegenerative disease. Considering that hyperphosphorylation of tau is a well-known marker of neurodegenerative disorders and that tau phosphorylation is one of the earliest cytoskeletal changes in AD and a critical step in the formation of neurofibrillary tangles (*Wischik et al., 1988*), we evaluated tau phosphorylation at Ser396 (pTau$^{Ser396}$) which has been strongly linked to AD progression (*Mondragón-Rodríguez et al., 2014*). Our Western blot analysis performed on the hippocampi of 3×Tg -AD and WT mice showed a marked increase of pTau$^{Ser396}$ in 6 M transgenic mice exposed to noise compared with age-matched not-exposed animals (*Figure 8A and B*; n = 4 animals/group; Student's t-test, AD-NE vs. AD-NN, p = 0.006), whereas no significant differences were observed between WT-NE and WT-NN groups (*Figure 8A and B*; n = 4 animals/group; Student's t-test, WT-NE vs. WT-NN mice p = 0.49).

We also studied TNF-α and IL-1β, which are well-known inflammatory markers playing a key role in neurodegenerative disease. Data of Western blot and ELISA analyses showed significant increases of both TNF-α (~90%) and IL-1β (7.50 ± 1.58 vs. 2.88 ± 0.47 pg/mg) in the hippocampus of 3×Tg -AD mice exposed to noise compared with age-matched not-exposed animals (*Figure 8C–E*) and no differences between WT-NN vs. WT-NE mice (*Figure 8C–E*; TNF-α, n = 4 animals/group; Student's t-test, AD-NE vs. AD-NN mice, p = 0.015; IL-1β, n = 3 animals/group; Student's t-test, AD-NE vs. AD-NN mice, p = 0.01).

To further characterize molecular underpinnings of hippocampal dysfunctions, we investigated oxidative stress, considering that redox imbalance is a well-known consequence of noise-induced hearing loss (*Fetoni et al., 2019*) and hippocampus is particularly prone to hearing loss-related oxidative stress (*Stebbings et al., 2016*; *Nadhimi and Llano, 2021*). Immunofluorescence analyses performed by using dihydroethidium (DHE) assay in coronal sections of both 6 M AD and WT mice showed an increase of reactive oxygen species (ROS) in DG (*Figure 9C–D, I*) and CA1 (*Figure 9G–H, I*) regions of the hippocampus of transgenic mice exposed to noise, compared with age-matched not-exposed animals. Fluorescence signal quantification showed a significant increase of fluorescence intensity specifically in AD-NE compared with AD-NN mice (*Figure 9I and J*; n = 3 animals/group; Student's t-test, CA1, AD-NE vs. AD-NN mice, p = 0.001; DG AD-NE vs. AD-NN mice, p = 0.008).

Furthermore, considering that increased ROS production can lead to lipid peroxidation (*Angelova and Abramov, 2018*), we assessed lipid peroxidative damage. Peroxidation of the membrane lipid

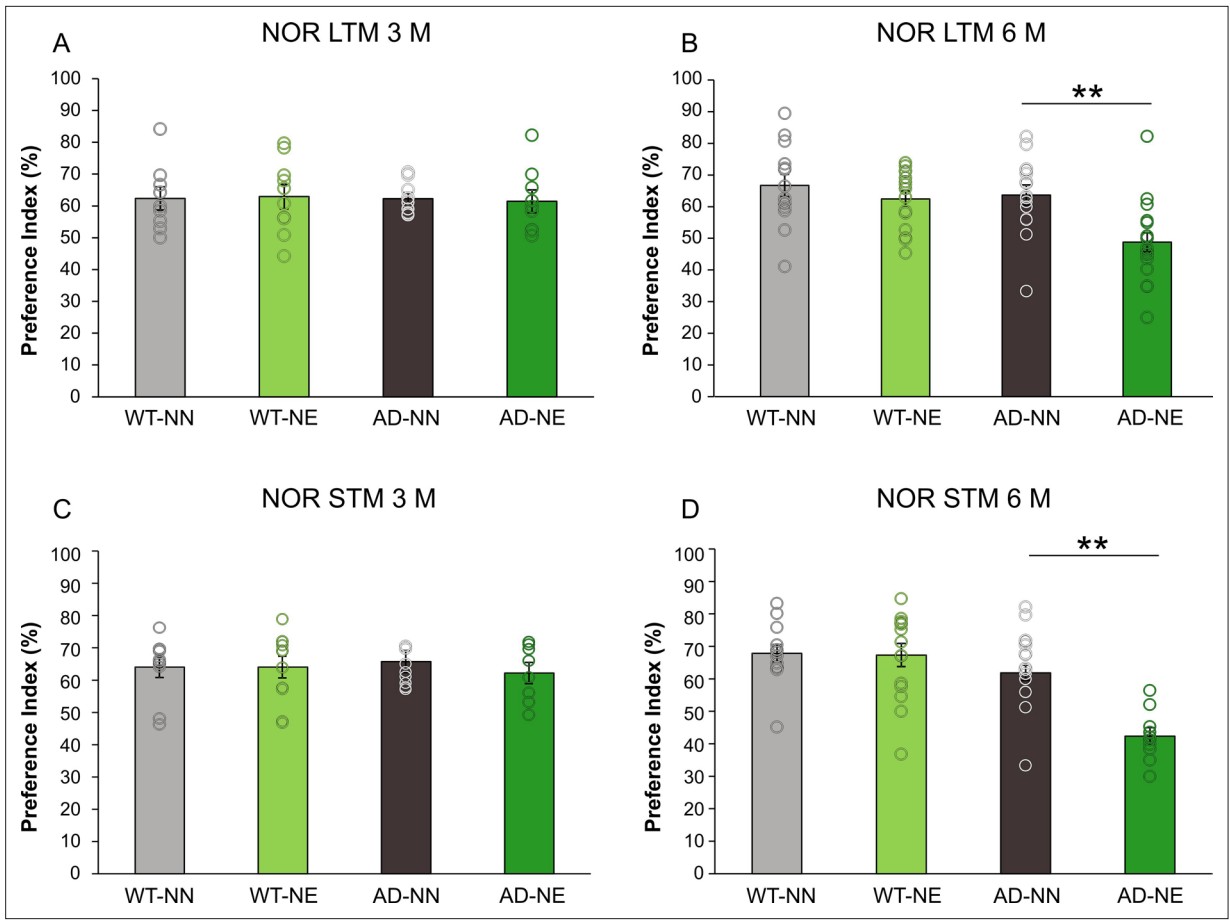

**Figure 7.** Sensory deprivation induced by noise accelerates learning and memory deficits in 3×Tg Alzheimer's disease (AD) mice. Graphs show preference index for the novel object in the novel object recognition (NOR), from both long-term memory (LTM; **A,B**) and short-term memory (STM; **C,D**) paradigms in NN and NE groups from wild-type (WT) and AD animals of 3 months of age (M) (A,C; WT-NN STM n = 10, LTM n = 9; WT-NE STM n = 11, LTM n = 10; AD-NN STM n = 8, LTM n = 9; AD-NE STM n = 8, LTM n = 9) and 6 M (B,D; WT-NN STM n = 13, LTM n = 14; WT-NE STM n = 15, LTM n = 15; AD-NN STM n = 14, LTM n = 15; AD-NE STM n = 10, LTM n = 18). Notably, memory performance decreased significantly in AD mice exposed to noise compared to age-matched not-exposed animals at 6 M, for both LTM (two-way ANOVA, Bonferroni post hoc test, p = 0.001) and STM (two-way ANOVA, Bonferroni post hoc test, p < 0.001) evaluations. Data are expressed as mean ± SEM. Asterisks indicate significant differences among groups (**p < 0.001).

The online version of this article includes the following figure supplement(s) for figure 7:

**Source data 1.** Numerical source data from *Figure 7*.

**Figure supplement 1.** Locomotor activity of wild-type (WT) and 3×Tg Alzheimer's disease (AD) mice of 3 months of age (M) and 6 M in the open field.

**Figure supplement 2.** Animal-by-animal analysis comparing auditory brainstem recording (ABR) measurements, novel object recognition (NOR) performance, and ACx spine density.

bilayer is one of the major sources of free radical-mediated injury that directly damages neurons causing increased membrane rigidity, decreased activity of membrane-bound enzymes, impairment of membrane receptors, and altered membrane permeability eventually leading to cell death (*Tekpli et al., 2013*). Of note, our results revealed an increase of 4-hydroxy-2-nonenal (4-HNE), a key marker of lipid peroxidation, in the hippocampus of 6 M 3×Tg -AD mice exposed to noise, compared to age-matched not-exposed mice (*Figure 10C–D and G–H*). This difference was significant in both CA1 and DG hippocampal regions, as confirmed by quantitative fluorescence signal analysis (*Figure 10I and J*; n = 3 animals/group; Student's t-test, CA1, AD-NE vs. AD-NN mice, p = 0.002; DG, AD-NE vs. AD-NN mice, p = 0.0001). No significant changes were found in WT-NE vs. WT-NN mice.

In order to confirm the increase of oxidative stress, we performed a dot blot to detect nitrotyrosine (NT), a marker of nitro-oxidative stress. Indeed, protein tyrosine nitration represents a prominent post-translational redox modification and it is associated with different diseases (*Ischiropoulos and*

**Table 1.** Exploration time for novel and familiar objects in both short- and long-term memory.

| Treatment | Short-term memory NOR | | Long-term memory NOR | |
|---|---|---|---|---|
| *3* M | *Novel obj.* | *Familiar obj.* | *Novel obj.* | *Familiar obj.* |
| WT-NN | 14 ± 2.1 s | 8 ± 1.3 s | 18.8 ± 4.2 s | 10.3 ± 1.2 s |
| WT-NE | 14.6 ± 1.5 s | 8.7 ± 1.4 s | 15.7 ± 3 s | 8.6 ± 1.3 s |
| AD-NN | 14.1 ± 1.2 s | 7.6 ± 1.1 s | 11.4 ± 1.9 s | 6.8 ± 0.9 s |
| AD-NE | 10.9 ± 1.1 s | 6.9 ± 1.1 s | 9.2 ± 0.9 s | 5.8 ± 0.8 s |
| *6* M | *Novel obj.* | *Familiar obj.* | *Novel obj.* | *Familiar obj.* |
| WT-NN | 19.7 ± 2.8 s | 8.7 ± 1.1 s | 15.2 ± 1.9 s | 7.3 ± 1 s |
| WT-NE | 16.3 ± 1.9 s | 7.6 ± 0.9 s | 17.5 ± 1.5 s | 10.9 ± 1.5 s |
| AD-NN | 13.1 ± 1.8 s | 7.7 ± 0.9 s | 11.4 ± 1.4 s | 6.3 ± 0.9 s |
| AD-NE | 6.8 ± 0.7 s | 9.4 ± 1 s | 7.1. ± 0.7 s | 7.2 ± 0.5 s |

*Beckman, 2003*). Dot blot analysis showed an increase of NT formation in hippocampal samples of 6 M AD-NE mice, compared to age-matched AD-NN animals (*Figure 10—figure supplement 1*; n = 4 animals/group; Student's t-test, AD-NE vs. AD-NN, p = 0.038).

Furthermore, considering the higher increase of oxidative stress in AD-NE mice, we looked for apoptosis markers, in order to investigate if altered redox status was associated with neuronal death. Western bot analysis revealed a significant increase of the pro-apoptotic Bax (n = 4 animals/group; Student's t-test, AD-NE vs. AD-NN mice, p = 0.004) and active Caspase-3 (n = 4 animals/group; Student's t-test, AD-NE vs. AD-NN mice, p = 0.024) proteins in hippocampi of AD-NE mice compared to age-matched not-exposed animals (*Figure 10—figure supplement 1*).

Finally, to study in depth redox status imbalance, we also investigated the endogenous antioxidant system, focusing on endogenous players of defense against free radical-induced damage, such as superoxide dismutase 2 (SOD2) and the inducible isoform of heme oxygenase-1 (HO-1). Our Western blot analysis revealed no changes in SOD2 expression in the hippocampus of noise-exposed animals compared to not-exposed groups, neither in 3×Tg AD nor in WT mice at 6 M (*Figure 11A and B*; n = 4 animals/group; Student's t-test, WT-NE vs. WT-NN mice, p = 0.81; AD-NE vs. AD-NN mice, p = 0.76). WT-NE mice showed increased HO-1 expression compared to WT-NN animals (*Figure 11C and D*; n = 4 animals/group; Student's t-test, p = 0.006), suggesting an endogenous antioxidant response activation to face a toxic insult. Conversely, no significant modulation of HO-1 expression was found in the hippocampus of 3×Tg -AD mice exposed to noise, compared to age-matched not-exposed animals (*Figure 11C and D*; n = 4 animals/group; Student's t-test, p = 0.93), likely due to an impaired ability of the endogenous antioxidant response to counteract oxidative stress and to restore redox balance.

## Discussion

Hearing impairment is known as a major clinical risk factor for cognitive decline (*Gates and Mills, 2005*; *Lin et al., 2011*; *Loughrey et al., 2018*; *Griffiths et al., 2020*; *Johnson et al., 2021*), with relevant clinical implications for dementia prevention, diagnosis, and treatment (*Dawes et al., 2015*; *Taljaard et al., 2016*; *Livingston et al., 2020*). However, the complex pathophysiological relation between hearing impairment and dementia remains to be fully defined.

The present research shows that auditory sensory deprivation induced by noise exposure in young mice (i.e., at 2 M, when AD phenotype is not manifested yet in 3×Tg AD mice) caused structural and functional changes in ACx. Notably, WT animals recovered over time, whereas 3×Tg -AD mice failed to rescue central damage associated with hearing loss, showing not only persistent synaptic and morphological alterations in the ACx, but also hippocampal dysfunction, increased tau phosphorylation, and anticipated memory deficits compared to the expected time-course of AD phenotype observed in age-matched 3×Tg -AD mice not exposed to noise. Neuroinflammation and redox imbalance were found in hippocampi of 6 M AD-NE mice that might be related to the reduced ability of the

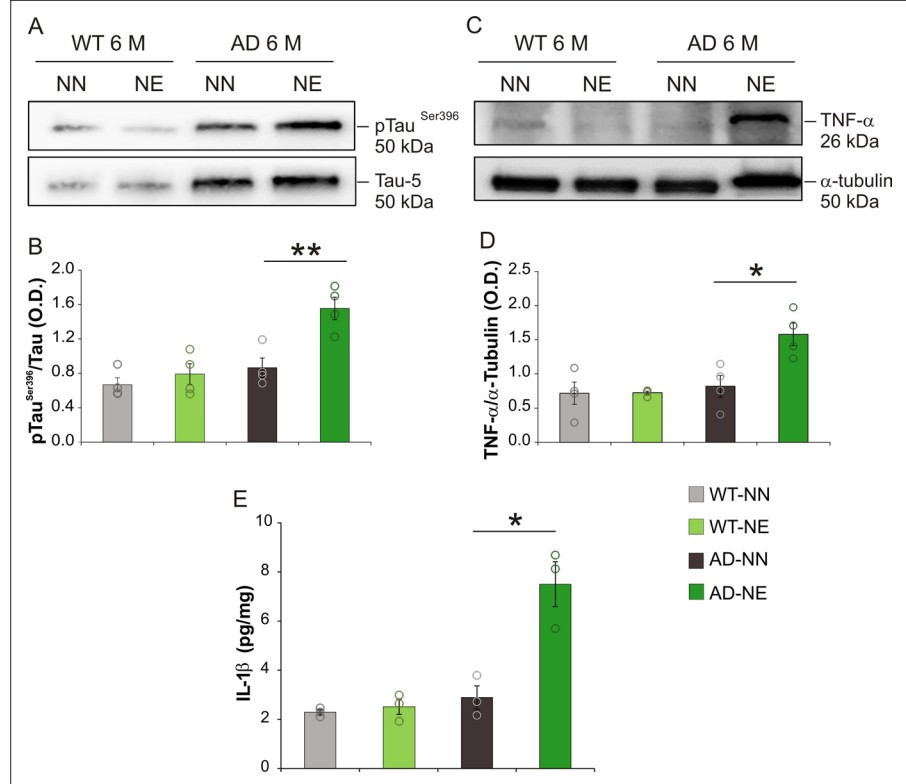

**Figure 8.** Increased tau phosphorylation and neuroinflammation in hippocampus of 3×Tg Alzheimer's disease (AD) mice. (A) Representative immunoblot revealing increased pTau$^{Ser396}$ in the hippocampus of 3×Tg -AD mice exposed to noise (6 months of age [M]) compared with age-matched not-exposed group. (B) Bar graph shows results of densitometric analyses on all samples (n = 4 mice for each group; Student's t-test, WT-NE vs. WT-NN, p = 0.64; AD-NE vs. AD-NN, p = 0.006) normalized to the corresponding total protein levels (Tau). (C) Representative Western immunoblot revealing increasing TNF-α expression in the hippocampus of AD exposed to noise (6 M) compared with age-matched not-exposed group. (D) Bar graph shows results of densitometric analyses on all samples (n = 4 mice for each group, Student's t-test; WT-NE vs. WT-NN, p = 0.95; AD-NE vs. AD-NN, p = 0.015) normalized to total protein levels (α-tubulin). Data are expressed as mean ± SEM. Asterisks indicate significant differences between groups (*p < 0.05; **p < 0.01). (E) Bar graph showing hippocampal IL-1β levels measured at 6 M in both NN and NE WT and AD mice (n = 3 mice for each group; Student's t-test, WT-NE vs. WT-NN, p = 0.60; AD-NE vs. AD-NN, p = 0.01). Data are expressed as mean ± SEM. Asterisks indicate significant differences among groups (*p < 0.05; **p < 0.01).

central structures to rescue noise-induced detrimental effects, making this experimental model of AD more vulnerable to central damage induced by hearing loss.

The experimental model of auditory sensory deprivation was achieved in both WT and 3×Tg -AD animals by exposing 2 M mice to repeated noise sessions, a paradigm used in our previous studies capable of inducing hearing loss, hair cell loss, and decreased transmission between inner hair cells and primary afferent fibers (*Fetoni et al., 2013*; *Paciello et al., 2018*).

In our study, no differences in baseline auditory thresholds were found between AD-NN and WT-NN animals at 3 and 6 M, consistent with previous data reporting alterations in cochlear functions starting from 9 M in other models of AD pathology (*Wang and Wu, 2015*; *O'Leary et al., 2017*).

The acoustic trauma caused permanent increase of auditory threshold of about 30–40 dB, spanning at all frequencies analyzed in both strains, estimated 1 and 4 months after the onset of trauma sessions (corresponding to 3 and 6 M of mouse age), indicating no differences in cochlear susceptibility to noise exposure between WT and 3×Tg -AD animals. By investigating the up-spread damage associated to noise-induced hearing loss, we found, consistent with our previous results (*Fetoni et al., 2015*; *Paciello et al., 2018*), both functional and morphological alterations 1 month after acoustic trauma. The spine loss observed in 3 M animals might be caused by deafferentation and activity-dependent remodeling of neuronal connectivity. Accordingly, our electrophysiological analyses on ACx brain

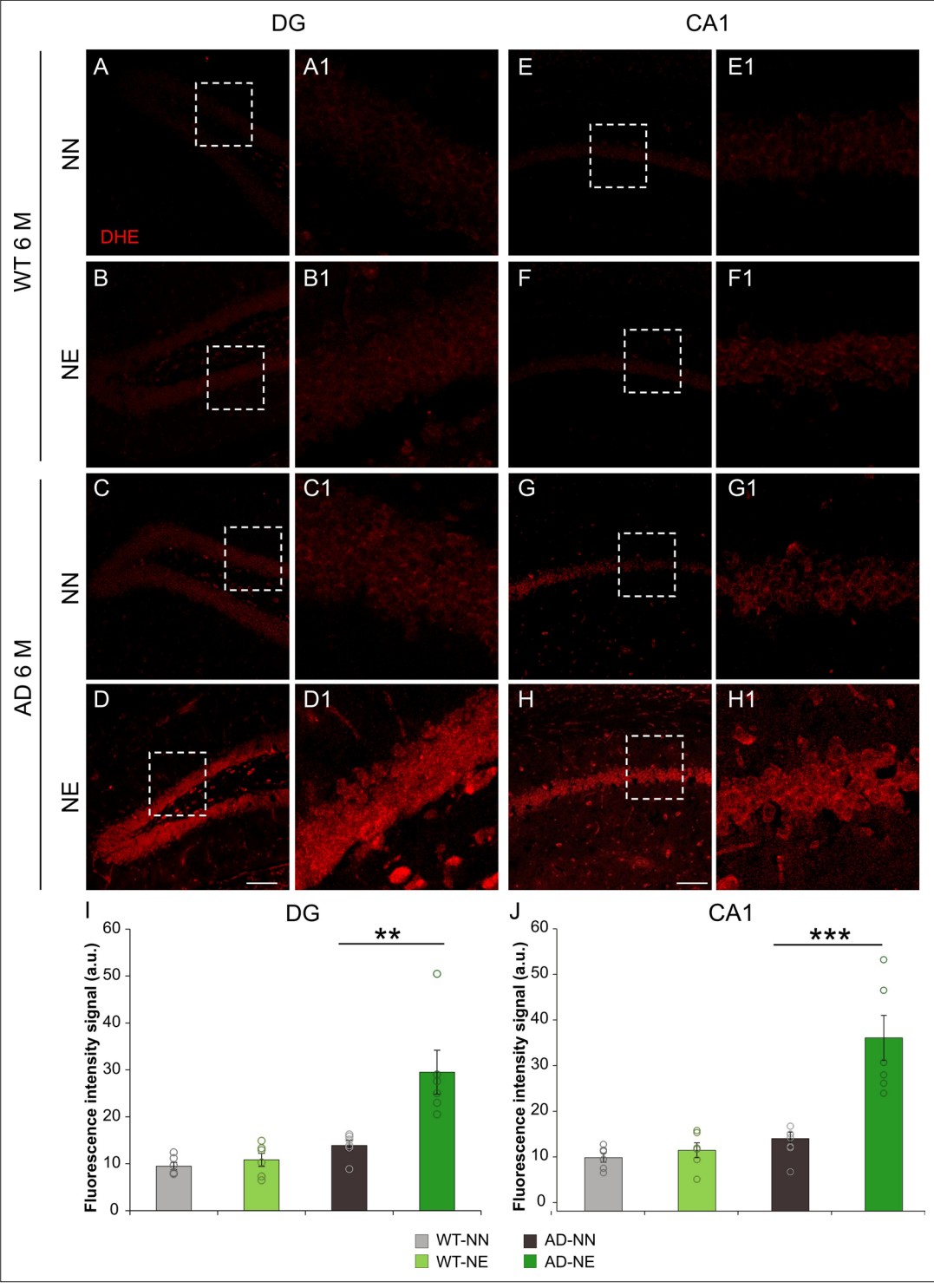

**Figure 9.** Noise induces increased reactive oxygen species (ROS) amount in the hippocampus of 3×Tg Alzheimer's disease (AD) mice. (A–H) Representative images of brain coronal sections stained with dihydroethidium (DHE) (red fluorescence) showing ROS amount in dentate gyrus (DG) (**A–D**) and CA1 (**E–H**) hippocampal regions of both not-exposed and noise-exposed wild-type (WT) and AD mice at 6 months of age (M). Dotted boxes in A–D and E–H refer to high magnifications showed in A1–D1 and E1–H1 respectively. (I–J) Bar graphs showing fluorescence intensity signal quantification in DG (**I**) and CA1 (**J**) in all groups. Data are expressed as mean ± SEM and are representative of three independent experiments from three animals/group. Scale bar: 100 μm. Asterisks indicate significant differences between groups (**p < 0.01; ***p < 0.001, Student's t-test).

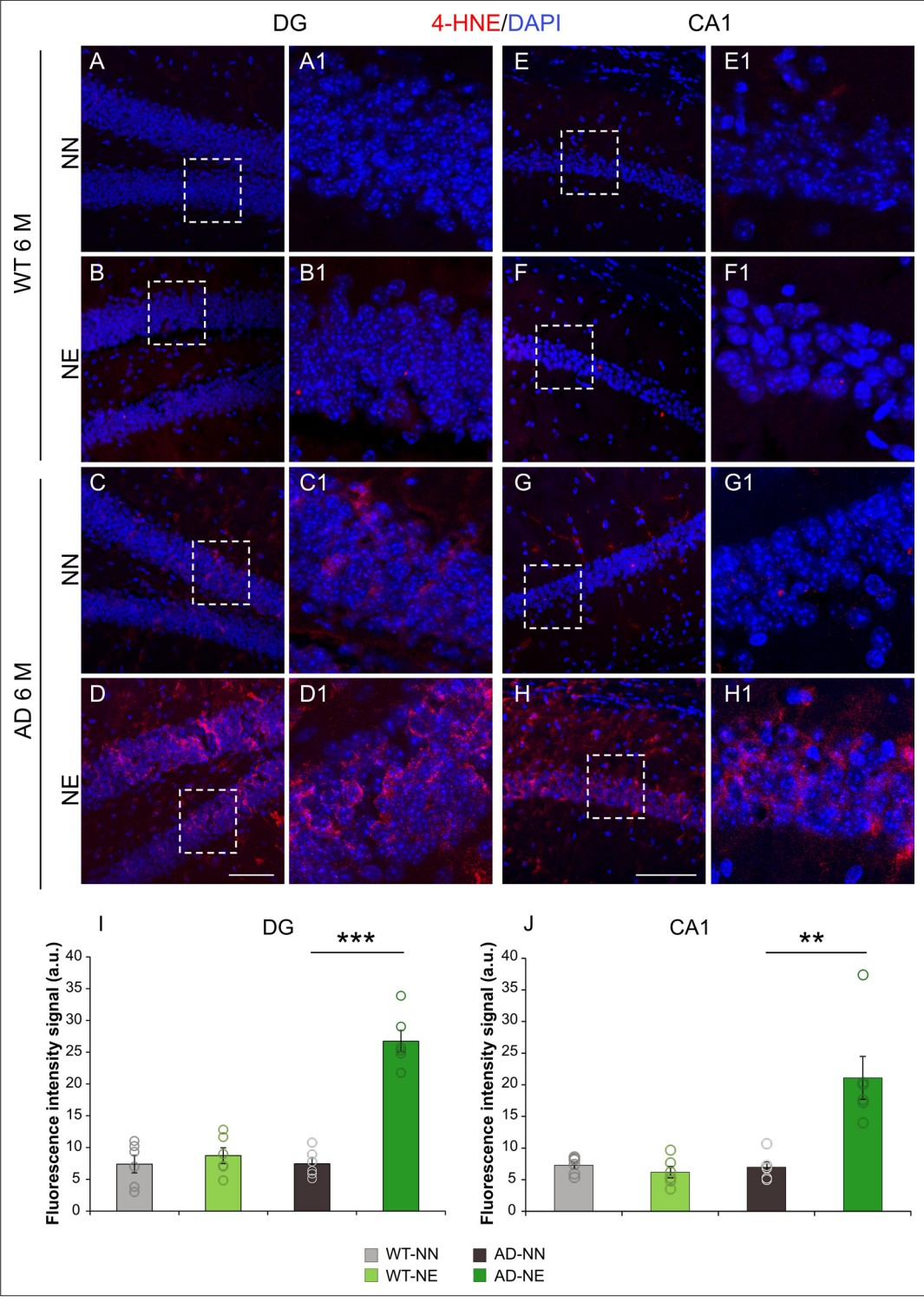

**Figure 10.** 3×Tg Alzheimer's disease (AD) mice exposed to noise show increased lipid peroxidation in the hippocampus. (A–H) Representative images of brain coronal sections stained with 4-hydroxy-2-nonenal (4-HNE) as a marker of lipid peroxidation (red fluorescence) and DAPI (blue fluorescence) to label cell nuclei in dentate gyrus (DG) (**A–D**) and CA1 (**E–H**) hippocampal regions of both not-exposed and noise-exposed wild-type (WT) and AD mice at 6 months of age (M). Dotted box in A–D and E–H refers to high magnifications showed in A1–D1 and E1–H1 respectively. (I–J) Bar graphs showing 4-HNE fluorescence intensity signal quantification in DG (**I**) and CA1 (**J**) in all groups. Data are expressed as mean ± SEM and are representative of three independent experiments from three animals/group. Scale bar: 100 μm. Asterisks indicate significant differences between groups (**p < 0.01; ***p

*Figure 10 continued on next page*

*Figure 10 continued*

< 0.001, Student's t-test).

The online version of this article includes the following figure supplement(s) for figure 10:

**Figure supplement 1.** Detection of hippocampal protein tyrosine nitration and increase of apoptotic markers.

slices revealed that noise-induced morphological changes were accompanied with decreased excitatory synaptic responses of neurons within layer II/III.

Interestingly, by analyzing the long-lasting effect of sensory deprivation (i.e., 4 months after noise exposure) in ACx, we found out that 6 M WT animals showed a recovery in both basal synaptic transmission and spine density in ACx, whereas 3×Tg -AD animals failed to recover synaptic and morphological dysfunctions associated with cochlear peripheral damage.

Moreover, we found that 6 M AD-NE mice showed decreased $pGluA1^{Ser845}$ at 6 M, suggesting a decreased AMPAR stabilization in the plasma membrane and potentially a decreased channel conductance in AD mice subjected to acoustic trauma. Such post-translational modification of GluA1 is known to stabilize glutamate receptor at dendrites (*Kessels et al., 2009*) and it is considered critical for sensory deprivation-induced homeostatic synaptic response and for experience-dependent synaptic plasticity (*He et al., 2009*; *Goel et al., 2011*). Whether decreased $pGluA1^{Ser845}$ could be a sign of frailty or impairment at glutamatergic synapses within ACx layer II/III needs to be elucidated in further studies. Our results showing a high vulnerability of auditory central structures to sensory deprivation in a transgenic model of AD are consistent with clinical evidence. Indeed, central auditory processing dysfunction is highly evident in patients with AD (*Idrizbegovic et al., 2011*), and pathological changes have also been found in the ascending auditory pathway (*Sinha et al., 1993*; *Parvizi et al., 2001*; *Baloyannis et al., 2009*) as well as in the ACx (*Lewis et al., 1987*; *Baloyannis et al., 2011*).

To substantiate the hypothesis that long-lasting alterations in ACx caused by noise-induced hearing loss would affect a crucial structure involved in memory, namely the hippocampus, we analyzed basal synaptic transmission at hippocampal Schaffer collateral-CA1 pyramidal neuron synapses as well as spine density of CA1 and DG neurons in 6 M animals. Interestingly, we found altered basal synaptic

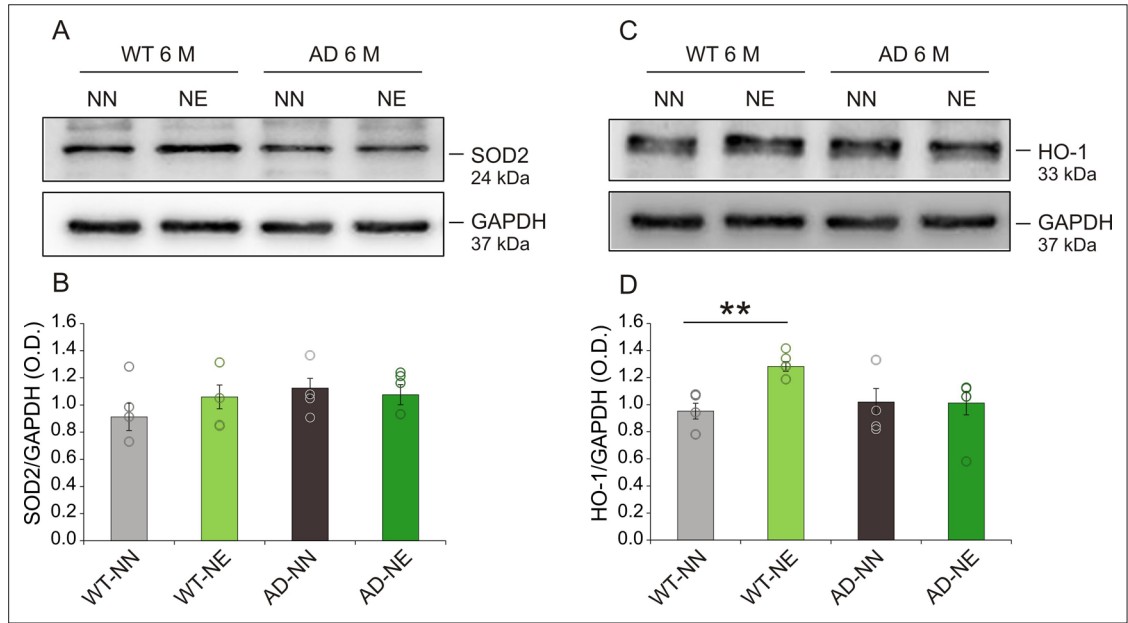

**Figure 11.** Expression of endogenous antioxidant enzymes in hippocampus after noise exposure. (A,C) Representative Western immunoblots revealing expression of superoxide dismutase 2 (SOD2) (**A**) and heme oxygenase-1 (HO-1) (**C**) in the hippocampus of both not-exposed (NN) and noise-exposed (NE) wild-type (WT) and Alzheimer's disease (AD) mice at 6 months of age (M). (B,D) Bar graphs showing results of densitometric analyses on all samples normalized to the corresponding total protein levels (GAPDH). No differences in SOD2 expression were found among groups (n = 4 mice for each group; Student's t-test, WT-NE vs. WT-NN, p = 0.81; AD-NE vs. AD-NN, p = 0.76), whereas a significant increase in HO-1 expression was found in hippocampi of WT animals exposed to noise (n = 4 mice for each group; Student's t-test, WT-NE vs. WT-NN, p = 0.0007; AD-NE vs. AD-NN, p = 0.95). Data are expressed as mean ± SEM. Asterisks indicate significant differences between groups (**p < 0.01).

transmission and reduced spine density only in the hippocampus of noise-exposed 3×Tg -AD mice, indicating that in this mouse model of AD the hippocampus is more susceptible to detrimental effects of hearing loss than WT mice.

It is known that hippocampus participates in processing of auditory information conveyed by the lemniscal and non-lemniscal paths (*Steward, 1976*; *Germroth et al., 1989*; *Moxon et al., 1999*; *Budinger and Scheich, 2009*; *Munoz-Lopez et al., 2010*; *Zhang et al., 2018*). Both of these pathways convey information from the cochlear nuclei to the hippocampus, which is critically involved in spatial learning tasks (*Adams et al., 2008*). On the other hand, the hippocampus can respond to acoustic as well as to visual and olfactory stimuli as it processes such information to create spatial memories (*Kemp and Manahan-Vaughan, 2008*; *André and Manahan-Vaughan, 2013*; *Dietz and Manahan-Vaughan, 2017*). Moreover, it has been reported that age-related hearing loss in an animal model of presbycusis is accompanied by extensive reorganization of plasticity-related neurotransmitter expression in ACx and hippocampus, and is associated with altered hippocampal synaptic plasticity, as well as memory impairments (*Beckmann et al., 2020*).

Although a certain degree of heterogeneity has been reported in 3×Tg -AD mouse model regarding the onset and progression of cognitive deficits, the majority of studies including ours indicate that 7–9 months is the time-window when cognitive deficits most often manifest (*Clinton et al., 2007*; *Martinez-Coria et al., 2010*; *Chakroborty et al., 2019*; *Joseph et al., 2019*; *Cocco et al., 2020*). Consistently, memory performance in 6 M 3×Tg -AD mice was comparable to that of WT mice. Of note, our study showed that sensory deprivation induced by noise accelerated cognitive decline in 3×Tg- AD mice, anticipating it at 6 M of age, as revealed by decreased LTM and STM recognition memory performance in NOR test.

Looking for a molecular mechanism underlying hippocampal functional and morphological alterations, we focused on common pathogenic markers shared by noise-induced hearing loss and neurodegenerative disease. Our Western blot and immunofluorescence analyses revealed increased pTau$^{Ser396}$ in the hippocampus of 6 M 3×Tg -AD mice exposed to noise in parallel with increased TNF-α and IL-1β, well-known markers of neuroinflammation, and increased oxidative stress, 4-HNE expression, and apoptotic markers in the hippocampus.

The interplay between oxidative stress and neuroinflammation is a common feature of AD pathology and production of oxidizing free radicals, including ROS and reactive nitrogen species, can be induced by increased cytokine production (*Naik and Dixit, 2011*). Moreover, several studies also showed that oxidative stress leads to increased tau phosphorylation in vitro (*Zhu et al., 2005*; *Su et al., 2010*) and peroxidative damage and 4-HNE expression facilitated aggregation of phosphorylated tau (*Pérez et al., 2000*), inducing tau hyperphosphorylation (*Gómez-Ramos et al., 2003*; *Liu et al., 2005*). Moreover, hyperphosphorylated tau accumulation in the hippocampus and neocortex and cognitive impairment have been observed as a consequence of chronic noise exposure in previous studies (*Cui et al., 2012*; *Park et al., 2018*). Our findings suggest that auditory sensory deprivation can exacerbate pathological molecular pathways of AD, accelerating cognitive decline.

Moreover, the increased expression of HO-1 in the hippocampus of WT animals exposed to noise can be considered as an endogenous response to prevent or counteract redox imbalance. Of relevance, we did not observe this HO-1 upregulation in AD-NE mice, suggesting an impaired ability of endogenous antioxidant system to face oxidative insult in AD mouse model. This hypothesis is consistent with previous studies demonstrating that HO can interact with APP family members resulting in HO inhibition. Accordingly, in APP mutant mice HO-1 as well as bilirubin levels were significantly decreased while oxidative stress-induced neurotoxicity was markedly increased (*Takahashi et al., 2000*). The observed changes in ROS/HO-1 expression are potentially responsible for the functional and structural alterations we found in the hippocampus of AD-NE mice. Here, we did not directly investigate the cause-effect relationship between oxidative stress and impaired synaptic plasticity, that we plan to address in a follow-up study. However, literature data allows us to hypothesize a role of oxidative stress, that is involved in various pathological states including several age-related neurodegenerative diseases, considering that neurons are seen as a crucial target of oxidative attacks (*Di Domenico et al., 2017*). Regarding SOD2, we did not find significant difference in protein expression among groups. However, considering the higher ROS and 4-HNE levels documented in the hippocampus of AD-NE mice, we cannot exclude changes in SOD2 enzymatic activity instead of protein expression per se.

Collectively, our data indicate that in 3×Tg -AD mice, central compensatory mechanisms to restore the up-spread damage induced by hearing loss are compromised, likely by early impairment of networks preceding the onset of the neurodegenerative pathology. As such, central damage-induced auditory sensory deprivation in the pre-symptomatic AD phase further compromises fragile networks, probably targeting common pathogenetic pathways, thereby accelerating onset and progression of AD phenotype.

In the context of high brain vulnerability and lower cognitive reserve, hearing loss is a critical risk factor for cognitive decline and AD onset. On a translational perspective, our data suggest that monitoring hearing and cognitive function in elderly people and attenuating hearing loss with rehabilitative devices could be effective tools for delaying and mitigating AD.

# Materials and methods

## Animals

Triple transgenic AD (3×Tg -AD) mice, harboring the Swedish human APP, presenilin M146V, and tauP301L mutations (*Oddo et al., 2003*) were used in this study, compared to (B6129SF2/J) WT mice (*Roddick et al., 2016*; *Ying et al., 2017*). Based on the mouse lifespan, the age of 3–6 months can be considered has a mature adult age (*Flurkey, 2007*) and according to previous studies, including ours (*Chakroborty et al., 2019*; *Joseph et al., 2019*; *Cocco et al., 2020*), 3×Tg -AD mice at the age of 2–3 months can be considered a model of preclinical AD, whereas the onset of AD phenotype manifests at 7–9 months. Sex differences in the prevalence, risk, and severity of AD as well as noise-induced hearing loss have been demonstrated in numerous clinical and animal studies (*Ferretti et al., 2018*; *Dumitrescu et al., 2019*; *Gür et al., 2019*; *Zhu et al., 2021*), with greater susceptibility to AD (*Laws et al., 2016*; *Jiao et al., 2016*; *Koran et al., 2017*; *Yang et al., 2018*) and lower susceptibility to noise-induced hearing loss (*Milon et al., 2018*; *Shuster et al., 2019*) in females comparing to males. Considering the neuroprotective role of estrogens in noise damage, to better understand if and how hearing loss induced by noise exposure affected cognitive functions, we focused our study on male animals.

The colonies were established in-house at the Animal Facility of the Università Cattolica from breeding pairs purchased from the Jackson Laboratory. For each strain, two experimental groups comprised animals submitted to noise trauma (AD-NE, n = 32; WT-NE, n = 33) and not-exposed animals (AD-NN, n = 42; WT-NN, n = 34). WT and 3×Tg -AD mice and, within each group, noise-exposed and not-exposed mice were housed separately, in cages containing from three to five animals.

## Noise exposure

3×Tg -AD and WT mice were exposed to repeated noise sessions at  2 M. The animals were placed in the anechoic room and exposed to a 100 dB SPL during 10 consecutive days for 60 min each day. The noise had a 10 kHz center frequency. As described previously (*Fetoni et al., 2013*; *Paciello et al., 2018*), the sound was generated by a waveform generator (LAG-120B, Audio Generator; Leader Electronics Corporation) and amplified by an audio amplifier (A-307R; Pioneer Electronics). The sound was presented in an open field by a dome tweeter (TW340 × 0; Audax) positioned at the center of the cage. Sound level was measured using a calibrated 1⁄4-inch microphone (model 7017; ACO Pacific) and a calibrated preamplifier (Acoustic Interface System; ACO Pacific) Larson Davis sound photometer (LD-831C) was used.

This paradigm of acoustic trauma of repeated and high intensity stimulation can be representative of dangerous exposure in humans, in whom exposure to intensities >85–87 dB are generally considered limit safe values (*ISO, 1990*).

## Auditory brainstem response recordings

ABRs were recorded at low, mid, and high frequencies to analyze hearing function in all experimental groups. Animals were mildly anesthetized (ketamine 50 mg/kg+ medetomidine 0.5 mg/kg, intraperitoneal injection) and placed in the anechoic room. Three stainless steel recording electrodes were subcutaneously inserted posterior to the tested pinna (active), vertex (reference), and contralateral pinna (ground). ABRs were collected using a computer-controlled TDT System 3 (Tucker-Davis Technologies, Alachua, FL) data acquisition system with real-time digital signal processing. Tone bursts ranging from

6 to 32 kHz (1 ms rise/fall time, 10 ms total duration, 20 /s repetition rate) were presented monaurally in an open field using a horn tweeter (Tucker-Davis Technologies). The responses were filtered (100–3000 Hz bandpass), digitized, and averaged across 512 discrete samples at each frequency-level combination. ABRs were measured at low (6 kHz), mid (12, 16, and 20 kHz), and high (24 and 32 kHz) frequencies. Thresholds were determined by decreasing tone intensity in 5 dB steps starting at 100 dB and decreasing to 0 dB or until a reliably scored ABR Wave I component was detected. Baseline ABRs were recorded bilaterally testing each ear separately to ensure no consistent left-right ear ABR asymmetry. After noise exposure, we recorded auditory potentials from the right ear only. Thus, data presented refers to right ear auditory thresholds.

Auditory thresholds were evaluated before, 1 month, and 4 months after acoustic trauma (*Figure 1*).

The neural transmission time for the auditory nerve and cochlear nucleus was evaluated by studying the latency of Waves I and II. The latency of the ABR components was defined as the time from the computer triggering of the earphone to the waveform positive peak, including a 0.3 ms acoustic transit time between the earphone and the animal pinnae (*Fetoni et al., 2013*; *Fetoni et al., 2016*; *Paciello et al., 2018*).

Data of noise-exposed animals were compared with age-matched not-exposed mice.

## Electrophysiology

Field recordings were performed on coronal slices (400-µm-thick) containing the hippocampus and the ACx as previously described (*Podda et al., 2008*; *Podda et al., 2016*; *Paciello et al., 2018*). Briefly, mice were anesthetized by isoflurane inhalation (Esteve) and decapitated. The brain was rapidly removed and placed in ice-cold cutting solution (in mM: 124 NaCl, 3.2 KCl, 1 $NaH_2PO_4$, 26 $NaHCO_3$, 2 $MgCl_2$, 1 $CaCl_2$, 10 glucose, 2 sodium pyruvate, and 0.6 ascorbic acid, bubbled with 95 % $O_2$-5% $CO_2$; pH 7.4). Slices were cut with a vibratome (VT1200S) and incubated in artificial cerebrospinal fluid (aCSF; in mM: 124 NaCl; 3.2 KCl; 1 $NaH_2PO_4$, 26 $NaHCO_3$, 1 $MgCl_2$, 2 $CaCl_2$, 10 glucose; 95 % $O_2$-5% $CO_2$; pH 7.4) at 32 °C for 60 min and then at room temperature (RT) until use.

Slices were transferred to a submerged recording chamber and continuously perfused with aCSF (flow rate: 1.5 ml/min). The bath temperature was maintained at 30–32°C with an in-line solution heater and temperature controller (TC-344B, Warner Instruments). Identification of slice subfields and electrode positioning were performed with 4× and 40× water immersion objectives on an upright microscope (BX5IWI, Olympus) and video observation (C3077-71 CCD camera, Hamamatsu Photonics).

All recordings were made using MultiClamp 700B amplifier (Molecular Devices). Data acquisition and stimulation protocols were performed with the Digidata 1440 A Series interface and pClamp 10 software (Molecular Devices). Data were filtered at 1 kHz, digitized at 10 kHz, and analyzed both online and offline. Field recordings were made using glass pipettes filled with aCSF (tip resistance 2–5 MΩ). fEPSPs were evoked in CA1 pyramidal neurons of hippocampus by stimulation of the Schaffer collateral and in pyramidal neurons of ACx layer II/III by stimulation of local connections using a concentric bipolar tungsten electrode (FHC Inc, Bowdoin, ME) connected to a stimulator.

I/O curves were obtained by afferent fiber stimulation at intensities ranging from 20 to 300 µA (in increments of 30 or 50 µA; stimulus rate of 1 pulse every 20 s).

## Golgi-Cox staining

Golgi-Cox staining was performed in both noise-exposed and not-exposed animals at 3 and 6 M of age to evaluate morphological features of hippocampal neurons (CA1 and DG regions) and ACx pyramidal neurons of layer II/III. Mouse brains were dissected and used for Golgi-Cox staining according to previously published protocol (*Fetoni et al., 2013*; *Paciello et al., 2018*; *Barbati et al., 2020*). Neurons were identified and selected only if the labeling was uniform and lacked any reaction precipitate, they were relatively isolated from neighboring impregnated neurons to avoid overlapping, the predominant plane of the dendritic arbors was parallel to the plane of the section, the dendritic arborizations were intact and visible as far as the most distal branches of apical and basal dendrites, and spines were clearly marked. We counted only spines that protruded laterally from the dendritic arborization, excluding those above or below the dendrite. In a blinded manner, apical and basal dendritic trees were separately counted and spine density was calculated along ~20 µm length of dendritic terminals. The stained sections were analyzed using Olympus BX63 microscope with a 100× oil-immersion objective lens.

## Memory test

Behavioral tests were carried out from 9 a.m. to 4 p.m. and data were analyzed using an automated video tracking system (Any-Maze). Recognition memory, both STM and LTM, was evaluated using the NOR paradigm. Both tests were divided into three sessions: habituation, training, and test. Briefly, for the LTM version, animals were familiarized on the first day for 10 min with the test arena (45 cm × 45 cm). On the second day (training session), they were allowed to explore two identical objects placed symmetrically in the arena for 5 min. On the third day (test session), a new object replaced one of the old objects and animals were allowed to explore for 5 min. For the short-term version of the test, which took place 24 hr after the long-term version, animals were habituated to the arena for 10 min on the first day, and underwent training and test, separated by 30 min, on the second day. Different couples of objects were used for the short-term and for the long-term paradigms. Preference index, calculated as the ratio between time spent exploring the novel object and time spent exploring both objects, was used to measure recognition memory. To exclude place preference in the test session, the position of the novel object was alternated on both sides of the box. Furthermore, object identity was counterbalanced across group, to exclude preference based on intrinsic object properties. The objects used were: lego bricks arranged in different shapes, glass bottles filled with clean bedding, pyramids, and spheres made of plastic. All objects were of almost the same size, and lego bricks were arranged in simple shapes (cubical or rectangular). Each object during each phase was taped to the arena, so that the animals couldn't displace them. After each test, the objects and the box were cleaned with 70 % ethanol solution. Mice exploring less than 10 s were excluded from the analysis.

Moreover, to ensure that deficits in locomotor activity, which could have hindered object recognition test results, were not induced by the treatment, animal behavior was evaluated in the open field test. Briefly, distance traveled during the habituation phase of the NOR test was calculated as an index of locomotor activity. Analyses were performed using Any-Maze. Both the experiments and the analyses were performed by researchers blind to treatments.

In order to find a relationship between memory performance and auditory sensory deprivation, a simple linear regression model was estimated by performing an animal-by-animal study comparing mean ABR threshold, NOR performance, and mean number of spines in ACx by using Statistica (Statsoft) software.

## Oxidative stress evaluation

To assess the oxidative damage in hippocampus, we used DHE staining and 4-HNE immunostaining. DHE and 4-HNE provided indications on production of the toxic-free radicals and oxidative degradation of lipids generated by the effect of oxidative stress, respectively. Brains from three/animals/group were quickly removed after transcardial perfusion with PBS 4 % and, subsequently, with paraformaldehyde and samples were fixed with 4 % paraformaldehyde in PBS at 4°C and pH 7.4 overnight. Immunofluorescence analysis was performed on 30-μm-thick coronal brain cryosections (Cryostat, SLEE Medical GmbH, Germany) containing the hippocampus from both 3×Tg -AD and WT mice exposed to noise or not (at 6 M).

For DHE staining, slices were incubated with 1 μM DHE (*Supplementary file 1*) in PBS for 30 min at 37 °C and then coverslipped with an antifade medium (FluoSave; *Supplementary file 1*). For 4-HNE immunostaining, the slides were incubated in a blocking solution containing 1 % fatty acid-free bovine serum albumin (BSA), 0.5 % Triton X-100, and 10 % normal goat serum in PBS for 1 hr at RT. The specimens were then incubated overnight at 4 °C with a solution containing rabbit monoclonal anti-4-HNE primary antibody (*Supplementary file 1*). At the end of the incubation, all slides were washed twice in PBS and incubated at RT for 2 hr, light-protected, in labeled conjugated goat anti-rabbit secondary antibody (*Supplementary file 1*). After another wash in PBS, samples were double-stained with DAPI (*Supplementary file 1*) for 20 min in the dark at RT. DAPI labeling was used to identify condensed cell nuclei. The slides were coverslipped with an antifade medium (FluoSave; *Supplementary file 1*). Images were taken by a confocal laser scanning microscope. Fluorescent images were obtained with a confocal laser microscope (Nikon Ti-E, Confocal Head A1 MP, Tokyo, Japan) with a 20× objective lens.

For all immunofluorescence analysis, a semi-quantitative measure of fluorescence signals was performed: fluorescence intensity of each area of interest, corresponding to hippocampus CA1 and DG regions was quantified with ImageJ (version 1.51 s).

## Western immunoblotting and dot blot

Total proteins were extracted from ACx or hippocampus of not-exposed and noise-exposed mice sacrificed at 6 M, using ice-cold RIPA buffer (*Supplementary file 1*) as reported previously (*Podda et al., 2016*). Protein lysates (30 μg) were loaded onto 10 % or 12 % Tris-glycine polyacrylamide gels for electrophoretic separation. Precision Plus Protein Dual Color Standards (*Supplementary file 1*) were used as molecular mass standards. Proteins were then transferred onto nitrocellulose membranes at 330 mA for 2 hr at 4 °C in transfer buffer containing 25 mM Tris, 192 mM glycine and 20 % methanol. Membranes were incubated for 1 hr with blocking buffer (5 % skim milk in TBST), and then incubated overnight at 4 °C with primary antibodies directed against one of the following proteins: pGluA1$^{Ser845}$, GluA1, PSD-95, pTau $^{Ser396}$, Tau-5, TNF-α,SOD2, HO-1, Caspase-3, Bax, and GAPDH, or α-tubulin (*Supplementary file 1*).

For dot blot 5 μl of lysates were spotted into a TBST pre-wetted nitrocellulose membrane. After draining, equal loading of protein amounts was then verified by staining the membrane with Ponceau S. Then, the membrane was blocked using skim milk in TBST for 90 min. Protein tyrosine nitration was detected using a specific antibody for NT (*Supplementary file 1*).

After three 10 min rinses in TBST, membranes were incubated for 2 hr at RT with HRP-conjugated secondary antibodies (*Supplementary file 1*). The membranes were then washed, and the bands were visualized with an enhanced chemiluminescence detection kit (GE Healthcare, United Kingdom). Protein expression was evaluated and documented using UVItec Cambridge Alliance.

## ELISA measurements

Hippocampi of 6 M WT and 3×Tg -AD mice (both not-exposed and noise-exposed) were collected and stored at −80 °C until further use. IL-1β levels were determined using commercially available ELISA kits (*Supplementary file 1*). The assay was performed according to the manufacturer's instructions on samples collected from three animals per group, and each sample was analyzed in duplicate.

## Statistical analyses

Sample sizes were chosen with adequate statistical power (0.8) according to results of prior pilot data sets or studies, including our own using similar methods or paradigms. Data were first tested for equal variance and normality (Shapiro-Wilk test) and then the appropriate statistical tests were chosen. The statistical tests used (one-way ANOVA, two-way ANOVA, three-way ANOVA, or Student's t-test, simple linear regression) are indicated in the main text and in the corresponding figure legends for each experiment. Post hoc multiple comparisons were performed with Bonferroni or Tukey correction (SigmaPlot 14.0 or Statistica, Statsoft). The level of significance was set at 0.05. Results are presented as mean ± SEM. Analyses were performed blinded.

# Acknowledgements

This work was supported by BRiC INAIL 2016-DiMEILA17, INAIL Bando BRIC 09. Università Cattolica del Sacro Cuore contributed to the funding of this research project and its publication (D1, D3.1 intramural funds). Confocal analysis was performed at the 'Labcemi' facility of the same University.

# Additional information

## Funding

| Funder | Grant reference number | Author |
|---|---|---|
| Istituto Nazionale per l'Assicurazione Contro Gli Infortuni sul Lavoro | BANDO BRIC 09 | Gaetano Paludetti Claudio Grassi |
| Istituto Nazionale per l'Assicurazione Contro Gli Infortuni sul Lavoro | BRiC INAIL 2016-DiMEILA17 | Gaetano Paludetti Claudio Grassi |

| Funder | Grant reference number | Author |
|---|---|---|

The funders had no role in study design, data collection and interpretation, or the decision to submit the work for publication.

### Author contributions
Fabiola Paciello, Conceptualization, Data curation, Formal analysis, Investigation, Writing - original draft; Marco Rinaudo, Data curation, Formal analysis, Investigation; Valentina Longo, Sara Cocco, Giulia Conforto, Anna Pisani, Formal analysis, Investigation; Maria Vittoria Podda, Data curation, Supervision, Validation, Writing – review and editing; Anna Rita Fetoni, Conceptualization, Data curation, Supervision, Validation, Writing – review and editing; Gaetano Paludetti, Funding acquisition, Supervision; Claudio Grassi, Conceptualization, Funding acquisition, Project administration, Supervision, Validation, Writing – review and editing

### Author ORCIDs
Fabiola Paciello ⬤ http://orcid.org/0000-0002-8473-8074
Valentina Longo ⬤ http://orcid.org/0000-0002-5763-8879
Anna Pisani ⬤ http://orcid.org/0000-0003-0661-5017
Maria Vittoria Podda ⬤ http://orcid.org/0000-0002-2779-8417
Anna Rita Fetoni ⬤ http://orcid.org/0000-0002-6955-7603
Claudio Grassi ⬤ http://orcid.org/0000-0001-7253-1685

### Ethics
Animal procedures were approved by the Ethics Committee of the Catholic University and were fully compliant with Italian (Ministry of Health guidelines, Legislative Decree No. 26/2014) and European Union (Directive No. 2010/63/UE) legislation on animal research (Prot. No. 289/2018-PR). All efforts were made to limit the number of animals used and to minimize their suffering. The animals were housed two per cage at a controlled temperature (22-23 °C) and constant humidity (60-75%), under a 12 h light/dark cycle, with food (Mucedola 4RF21, Italy) and water ad libitum.

### Decision letter and Author response
Decision letter https://doi.org/10.7554/eLife.70908.sa1
Author response https://doi.org/10.7554/eLife.70908.sa2

## Additional files

### Supplementary files
• Supplementary file 1. List of reagents and antibodies used.
• Transparent reporting form
• Source data 1. Source data for western blots.

### Data availability
Source data files have been provided for ABR data in Figures 2, Field excitatory post-synaptic potential (fEPSP) data in Figure 3 and Figure 5, Spine density data in Figure 4 and Behavioral analysis data for Figure 7 and Figure 7—supplement 1. Original western blot bands from Figure 3, 4, 6, 8 and 11 have been provided.

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
