## [Decision Letter]

**Acceptance summary:**

This paper addresses an important question: How hearing impairment increases the risk of cognitive decline and dementia. Using an experimental mouse model of familial Alzheimer's disease, the authors show that noise-induced hearing impairment enhances a range of functional and structural synaptic deficits in the hippocampus and auditory cortex and accelerates memory dysfunction in Alzheimer's model mice. These results provide a potential explanation for the link between impaired hearing and dementia.

**Decision letter after peer review:**

Thank you for submitting your article "Auditory sensory deprivation induced by noise exposure exacerbates cognitive decline and hippocampal dysfunction in a mouse model of Alzheimer's Disease" for consideration by *eLife*. Your article has been reviewed by 3 peer reviewers, one of whom is a member of our Board of Reviewing Editors, and the evaluation has been overseen by Barbara Shinn-Cunningham as the Senior Editor. The reviewers have opted to remain anonymous.

Essential revisions:

(1) The ABR analysis did not include important information, especially which wave or waves were used for determining thresholds. This can affect the interpretation of results because changes in specific ABR waves could indicate where along the auditory pathway the noise-induced hearing loss occurs. For instance, a threshold shift in the later waves could not be simply attributed to cochlear injury, as suggested in the manuscript. In fact, the study of noise induced hearing loss in C57BL6 mice, cited above, reported that physiological effects were not necessarily associated with peripheral injury.

In addition, the methods state that the tone bursts were "…presented monaurally in an open field" and then states, "ABRs were assessed bilaterally in all animals…". Are the thresholds presented in Figure 2 an average of ABR thresholds for each ear?

(2) There is a disconnect between the auditory cortex physiology results and the behavior. The memory task does not rely on auditory processing or auditory plasticity. Therefore, the relationship between auditory cortex measures and an AD-associated behavioral phenotype should be addressed.

(3) The difference in ROS is dramatic between control and sensory-deprived AD mice. Could the authors confirm these findings by other tools? Does such a dramatic increase in ROS cause apoptosis of neurons, in addition to the reduction in spine density?

(4) Changes in ROS and in HO1 expression are only correlative and not linked to functional and structural impairments observed in AD model mice. In the previous studies, the authors show that cochlear impairments can be reversed by Q10. A possible causal link between oxidative stress and impairments in CA3-CA1 synaptic transmission / spine density can be tested by local injections of antioxidants to the hippocampus. Otherwise, these results remain purely correlative and do not imply that it is ROS that impairs homeostatic capabilities of hippocampus and auditory cortex in AD model.

(5) The study used only male mice, but did not provide a rationale for doing so. There is a broad literature that indicates sex-specific differences in age-related hearing loss, noise-induced hearing loss, and progression of AD /cognitive decline. Therefore, it would be particularly important for a study of this sort to consider sex as a critical biological variable.

(6) There is a potential concern regarding the statistical approach: For example,

In Figure 4, the groups in Figure 4C and 4D appear to have been analyzed using two separate ANOVAs. Since the dependent variable (i.e., number of spines) is the same for the different groups/conditions, one ANOVA should be used to establish whether or not a main effect is observed.

In Figure 3D, individual data points between Ctrl and Noise are largely overlapping, so significance may be influenced by the 'outlier' in the Ctrl group. Please clarify, and also if all assumptions for a Student's t-test are met here.

(7) In Figure 1, it seems that a large number of mice was included in this study (74 3xTg and 67 WT) but it is difficult to understand how these mice were distributed to individual experiments. Do the authors have individual mice that underwent all (or most) of the experimental steps, and if so an analysis on an animal-by-animal basis would be appreciated.

(8) More information regarding animals would be helpful, e.g. were mutant and wild-type mice housed together? Were any mice single housed? This may affect behavior. Were experimenters blinded for the behavioral experiments?

(9) It should be discussed / justified more explicitly how well the noise-induced hearing loss represents the type of midlife hearing loss that is present in humans.*Reviewer #1:*

The manuscript by Paciello et al., addresses an important question: How familial Alzheimer's disease (AD) mutations affect the response of the auditory cortex and the hippocampus to hearing loss. The authors use a paradigm of noise exposure that induces hearing loss by hair cell loss and by decreased synaptic transmission between inner hair cells and primary afferent fibers, as has been previously shown by the authors (Fetoni et al., 2013; Paciello et al., 2018). While short-term (1-month) effects of hearing loss have been previously established, the current manuscript is focused on the long-term (4-months) effects of hearing loss in WT and AD model. The authors demonstrate that hearing loss and its dynamics is similar between WT and AD model mice in response to noise exposure. However, the CNS response of AD mice to hearing loss is different in comparison to WT mice. WT mice show reversible changes in basal synaptic transmission and spine density in response to noise exposure in both, the auditory cortex and the hippocampus. In contrast, AD model mice show decreased basal synaptic transmission and spine density in the same brain structures 4 months after the noise exposure. Moreover, long-term hearing loss in AD, but not in WT mice, impaired performance of AD mice in novel object recognition task at the presymptomatic disease stage. Finally, the authors show that hearing loss increased tau phosphorylation, pro-inflammatory markers, ROS levels and reduced HO-1 expression. This work strongly suggests that familial AD mutations impair the compensatory system of the auditory cortex and the hippocampus, leading to long-term synaptic impairments. While electrophysiological and spine density measurements are well performed, the authors need to make a better link between the molecular changes they describe and the functional impairments induced by sensory deprivation in AD model.

*Reviewer #2:*

This study addresses the epidemiological link between adult induced hearing loss and cognitive decline associated with a genetic model of Alzheimer's disease. The study evaluates the structural and functional changes in both the auditory cortex and hippocampus following noise-induced hearing loss in adult mice. Their results suggest that noise-induced hearing loss caused long-lasting synaptic and morphological changes to auditory cortex neurons, but only in the mouse model of AD (i.e., 3-Tg-AD mice). 3-Tg-AD mice with noise-induced hearing loss also displayed hippocampal dysfunction (e.g. elevated tau-phosphorylation, elevated neuroinflammation, altered oxidative stress levels) and poorer memory (e.g., novel object recognition). The strength of the study is the breadth of assessments used which provide useful information about the additive effect of hearing loss in animals with Alzheimer's-associated genetic deficits. However, the study has weaknesses that undermine the interpretation and impact of the results. For instance, the control animals display age-related hearing loss at a fairly early stage of mouse development (possibly due to the C57BL6 background), the ABR analysis is not carried out rigorously, and there is no relationship between the auditory cortex findings and the behavioral assay.

*Reviewer #3:*

In this paper the authors use a range of tools to address the effects of hearing impairment on neuronal function and behavior in the 3xTg model of AD. They report that in this specific model noise-induced hearing loss produces persistent functional (ie, fEPSP) and structural (ie, spine numbers, pGluA1, PSD95) impairments not only in the primary auditory cortex but also more remotely in the hippocampus, also leading to an earlier onset of memory impairment. They also report that these deficits correlate with an increase in pTau, TNFalpha and Il-1beta as well as increased levels of reactive oxygen species and lipid peroxidation.

Understanding the link between hearing loss and dementia is timely and important, however I feel more focus on investigating the key mechanisms that explain their results could improve the paper.

In Figure 1, it seems that a large number of mice was included in this study (74 3xTg and 67 WT) but it is difficult to understand how these mice were distributed to individual experiments. Do the authors have individual mice that underwent all (or most) of the experimental steps, and if so an analysis on an animal-by-animal basis would be appreciated.

In Figure 3D, individual data points between Ctrl and Noise are largely overlapping, and I am worried that significance is influenced by the 'outlier' in the Ctrl group. Please clarify, and also if all assumptions for a Student's t-test are met here.

More information regarding animals would be helpful, e.g. were mutant and wild-type mice housed together? Were also only male wild type mice used? This must be stated clearly. Were any mice single housed? This may affect behavior. Were experimenters blinded for the behavioral experiments?

I was wondering how well the noise-induced hearing loss represents the type of midlife hearing loss that is present in humans, and I feel this should be justified more explicitly.

---

## [Author Response]

Essential revisions:(1) The ABR analysis did not include important information, especially which wave or waves were used for determining thresholds. This can affect the interpretation of results because changes in specific ABR waves could indicate where along the auditory pathway the noise-induced hearing loss occurs. For instance, a threshold shift in the later waves could not be simply attributed to cochlear injury, as suggested in the manuscript. In fact, the study of noise induced hearing loss in C57BL6 mice, cited above, reported that physiological effects were not necessarily associated with peripheral injury.

As requested, more information on ABR procedure has been added in the revised manuscript. Specifically, hearing threshold levels were determined as the stimulus intensity at which a Wave I peak could be identified. We agree with the Reviewer that the later waves do not reflect cochlear injury. In fact, the mouse ABR is composed of four components, reflecting neural activity chiefly from the auditory nerve (Wave I), the cochlear nucleus (Wave II), the superior olivary complex (Wave III), and the lateral lemniscus and/or inferior colliculus (Wave IV) (Alvarado et al., Neurosci Res 2012; Scimemi et al., Acta Oto Ital 2014, Willot, Handbook of Mouse Auditory Research: From Behavior to Molecular Biology: Taylor and Francis, 2001). To measure auditory thresholds we used Wave I, that in mice is the largest and usually the last wave to disappear as the sound stimulus decreases (Willot, Handbook of Mouse Auditory Research: From Behavior to Molecular Biology: Taylor and Francis, 2001). It has been documented that Wave I is affected by noise exposure in several models and it correlated with damage to inner hair cells and loss of auditory nerve fibers (Kujawa and Liberman, J Neurosci 2009; Lavinsky et al., PLOS Genet 2015; Mehraei et al., J Neurosci 2016; Jongkamonwiwat et al., Cell Rep 2020). In mice most studies use Wave I to determine thresholds (Willot, Handbook of Mouse Auditory Research: From Behavior to Molecular Biology: Taylor and Francis, 2001). Nonetheless, in the revised manuscript we added the analysis of Wave I and Wave II latency-intensity curves (reflecting the speed of neural transmission) to provide further electrophysiological data supporting noise-induced cochlear damage.

In addition, the methods state that the tone bursts were "…presented monaurally in an open field" and then states, "ABRs were assessed bilaterally in all animals…". Are the thresholds presented in Figure 2 an average of ABR thresholds for each ear?

In the revised manuscript we clarified the procedure used (see pages 20-21, lines 483-488). Specifically, to verify normal hearing before noise exposure, ABRs were recorded bilaterally to ensure no consistent left-right ear ABR asymmetry due to ear pathologies. After noise exposure, we do not expect inter-aural differences, considering that acoustic trauma delivered in open field induces bilateral and symmetric hearing loss. Thus the ABRs were recorded from the right ear only to increase the efficiency of data acquisition.

(2) There is a disconnect between the auditory cortex physiology results and the behavior. The memory task does not rely on auditory processing or auditory plasticity. Therefore, the relationship between auditory cortex measures and an AD-associated behavioral phenotype should be addressed.

To further investigate connections between alterations of the auditory cortex physiology and AD-associated behavioral phenotype, we performed an animal by animal analysis. Our data revealed no correlation between hearing thresholds and memory performance in NOR test. However, in noise exposed AD mice a high correlation between spine loss in auditory cortex and memory performance in NOR test was found (r^2^ = 0.96; P = 0.019; see page10, lines 226-239 and Figure 7- supplement 2). These results suggest a correlation between structural plasticity of the auditory cortex and memory performance. Literature data suggest a relationship between auditory processes and cognitive functions (Griffiths et al., Neuron 2020; Johnson et al., Brain 2020; Slade et al., Trends Neurosci 2020); nevertheless, more in depth analyses of the anatomical substrates linking alterations of auditory cortex plasticity and neural circuits underlying memory deserve further investigations.

(3) The difference in ROS is dramatic between control and sensory-deprived AD mice. Could the authors confirm these findings by other tools? Does such a dramatic increase in ROS cause apoptosis of neurons, in addition to the reduction in spine density?

To address these issues we performed additional Western blot experiments further investigating oxidative stress and apoptosis.

Specifically, our analyses in the hippocampus of 3×Tg-AD mice revealed an increase of Nitrotyrosine in transgenic mice exposed to noise, reflecting an enhancement of reactive nitrogen species (RNS) formation and confirming oxidative/nitrosative stress in AD mice exposed to noise. Moreover, increased expression of active Caspase-3 and the pro-apoptotic protein Bax were observed, suggesting activation of apoptotic pathways in the AD mice exposed to noise.

The results of these experiments are reported at pages 12-13, lines 287-298 and shown in Figure 10- supplement 1.

(4) Changes in ROS and in HO1 expression are only correlative and not linked to functional and structural impairments observed in AD model mice. In the previous studies, the authors show that cochlear impairments can be reversed by Q10. A possible causal link between oxidative stress and impairments in CA3-CA1 synaptic transmission / spine density can be tested by local injections of antioxidants to the hippocampus. Otherwise, these results remain purely correlative and do not imply that it is ROS that impairs homeostatic capabilities of hippocampus and auditory cortex in AD model.

We agree with the reviewer that additional experiments would be required to unequivocally demonstrate the cause-effect relationship between changes in ROS/HO1 expression and functional/structural impairments observed in AD model mice. However, we would like to draw the reviewer’s attention on the fact that, according to Italian laws, new experiments including local injection of antioxidants to the hippocampus require a specific approval from the Ethics Committee of our University followed by additional authorization by the Italian Ministry of Health. These administrative procedures require many months, that would delay much the submission of our manuscript revision. We would appreciate very much if this Reviewer accepted that such control experiment is postponed to a follow-up study. In the revised manuscript, we acknowledged that in our experimental model changes in ROS/HO1 expression are potentially responsible for functional and structural impairments observed in AD model mice and that further experimental evidence is required to establish a cause-effect relationship (see page 17, lines 414-420).

(5) The study used only male mice, but did not provide a rationale for doing so. There is a broad literature that indicates sex-specific differences in age-related hearing loss, noise-induced hearing loss, and progression of AD /cognitive decline. Therefore, it would be particularly important for a study of this sort to consider sex as a critical biological variable.

We agree with the Reviewers on the crucial role of gender differences. We have considered this potential criticism in planning our experiments. A rationale for the use of male animals has been added to the revised version of the manuscript (see page 19, lines 444-451). In fact, sex differences in the prevalence, risk and severity of AD have been demonstrated in numerous clinical and epidemiological studies (Zhu et al., Cell and Mol Life Sci 2021). Animal research with mouse models of AD also supports the greater susceptibility to AD in females (Koran et al., Brain Imaging Behav 2017; Laws et al., Curr Opin Psychiatry 2018;Yang et al., Neurosci. Bull 2018).

On the other hand, sexual hormones can influence noise-induced hearing loss in rodents (Milon et al., Biol Sex Differ 2018). Specifically, estrogens may have neuroprotective function in the inner ear (Meltser et al., J Clin Invest 2008; Shuster et al., J Acoust Soc Am 2019; Delhez et al., Cell Mol Life Sci 2020).

Taken together, females are more susceptible to AD pathology but they are less susceptible to noise-induced hearing loss. Based on this evidence and taken into account that male mice are more susceptible to noise-induced cochlear damage, we opted to focus our study on male animals.

(6) There is a potential concern regarding the statistical approach: For example,In Figure 4, the groups in Figure 4C and 4D appear to have been analyzed using two separate ANOVAs. Since the dependent variable (i.e., number of spines) is the same for the different groups/conditions, one ANOVA should be used to establish whether or not a main effect is observed.

A two-way ANOVA comparing all groups has been performed. Statistical results (F values and level of significance) have been added to the revised manuscript (see page 7, lines 151-152 and 155-156).

In Figure 3D, individual data points between Ctrl and Noise are largely overlapping, so significance may be influenced by the 'outlier' in the Ctrl group. Please clarify, and also if all assumptions for a Student's t-test are met here.

To identify outliers in our data set, we used the Outlier calculator in GraphPad Prism Software. This analysis identified no outliers from the curve fit and, therefore, no records were excluded from data analysis. However, we checked if statistical significance was affected by the highest data point in the Ctrl group and, after its removal, the level of significance was still > 0.05.

Statistical analysis has been revised and two-way ANOVA has been performed rather than Student’s t-test (see page 6, lines 129-130 and 132-133).

(7) In Figure 1, it seems that a large number of mice was included in this study (74 3xTg and 67 WT) but it is difficult to understand how these mice were distributed to individual experiments. Do the authors have individual mice that underwent all (or most) of the experimental steps, and if so an analysis on an animal-by-animal basis would be appreciated.

Sample size used for different experimental procedure is specified in each figure legend. Several animals underwent more than one experimental procedure (i.e., ABR, NOR, spine density evaluations). In the revised manuscript, we identified the subgroup of animals undergoing NOR test, ABR measurements and spine density and we performed an additional linear regression analysis to evaluate correlation index among these variables. Results of animal-by-animal analysis have been added in the revised manuscript (see page 10, lines 226-239) and shown in Figure 7- supplement 2.

(8) More information regarding animals would be helpful, e.g. were mutant and wild-type mice housed together? Were any mice single housed? This may affect behavior. Were experimenters blinded for the behavioral experiments?

WT and AD mice and, within each group, noise-exposed and no-noise mice, were housed separately, in cages containing from 3 to 5 animals (see page 19, lines 455-456).

(9) It should be discussed / justified more explicitly how well the noise-induced hearing loss represents the type of midlife hearing loss that is present in humans.

Noise-induced hearing loss caused by single exposure to very loud sounds or repeated exposures to lower intensity sounds over an extended period is a major source of hearing disability worldwide. Moreover, noise for occupational or leisure exposures is considered one of the major risk factors for hearing loss in midlife (Nelson et al., 2005; Dobie, 2008; European Agency for Safety and Health at Work, 2002; Lancet committee 2020). Furthermore, in experimental studies, noise-induced hearing loss represents a reliable and replicable model of sensorineural hearing loss.

Our paradigm of acoustic trauma of repeated and high intensity stimulation can be representative of dangerous exposure in humans, in whom exposure to intensities > 85-87 dB are generally considered limit safe values (ISO 1990; https://www.who.int; https://www.cdc.gov/niosh/topics/noise/default.html). Furthermore, based on the mouse lifespan, the age of 3-6 months can be considered has a mature adult age (https://www.jax.org/news-and-insights/jax-blog/2017/november/when-are-mice-considered-old) and considering that 3×Tg-AD mice at the age of 2-3 months can be considered a model of preclinical AD, whereas the onset of AD phenotype manifests at 7 to 9 months, we focused our study on 6 months of age animals. Details have been added in the revised manuscript (see page 19, lines 440-441 and page 20, lines 468-470).